

# *A Tale of Two Dust Storms*: Analysis of a Complex Dust Event in the Middle East

Steven D. Miller[1], Louie Grasso[1], Qijing Bian[2], Sonia Kreidenweis[2], Jack Dostalek[1], Jeremy Solbrig[1], Jennifer Bukowski[2], Susan C. van den Heever[2], Yi Wang[3], Xiaoguang Xu[3,4], Jun Wang[3], Annette Walker[5], Ting-Chi Wu[1], Milija Zupanski[1], Christine Chiu[2], and Jeffrey Reid[5]

[1] Cooperative Institute for Research in the Atmosphere, Colorado State University, Fort Collins, CO
[2] Department of Atmospheric Science, Colorado State University, Fort Collins, CO
[3] Department of Chemical and Biochemical Engineering, University of Iowa, Iowa City, IA
[4] Now at Joint Center for Earth Systems Technology, University of Maryland at Baltimore County, MD
[5] Naval Research Laboratory, Monterey, CA

*Correspondence to*: Steven D. Miller (Steven.Miller@colostate.edu)

**Abstract.** Lofted mineral dust over data-sparse regions presents considerable challenges to satellite-based remote sensing methods and numerical weather prediction alike. The Southwest Asia domain is replete with such examples, with its diverse array of dust sources, dust mineralogy, and meteorologically-driven lofting mechanisms on multiple spatial and temporal scales. A microcosm of these challenges occurred over 3-4 August 2016 when two dust plumes, one lofted within an inland dry air mass and another embedded within a moist air mass, met over the Southern Arabian Peninsula. Whereas conventional infrared-based techniques readily detected the dry air mass dust plume, they experienced marked difficulties in detecting the moist air mass dust plume, which only became apparent when visible reflectance revealed it crossing over an adjacent dark water background. In combining information from numerical modelling, multi-satellite/multi-sensor observations of lofted dust and moisture profiles, and idealized radiative transfer simulations, we develop a better understanding of the environmental controls of this event, characterizing the sensitivity of infrared-based dust detection to column water vapor, dust vertical extent, and dust optical properties. Differences in assumptions of dust complex refractive index translate to variations in the sign and magnitude of the split-window brightness temperature difference commonly used for detecting mineral dust. A multi-sensor technique for mitigating the radiative masking effects of water vapor via modulation of the split-window dust-detection threshold, predicated on idealized simulations tied to these driving factors, is proposed and demonstrated. The new technique, indexed to independent-sensor description of the surface-to-500 mb atmospheric column moisture, reveals parts of the missing dust plume embedded in the moist air mass, with best performance over land surfaces.

## 1 Introduction

Mineral dust poses unique and significant challenges to U.S. Navy operations in terms of its effects on visibility, electronics performance, and electro-optical signal propagation. The coastal (littoral) zones represent a unique challenge for analyzing and predicting the evolution of aerosol distributions and properties. The land/sea interface (surface and air mass



discontinuities) gives rise to highly dynamic and diurnally-varying flow patterns which redistribute dust horizontally and
vertically, and through changes in humidity, modulate its optical properties on fine spatial and temporal scales [van den
Heever et al., 2006; Igel et al., 2018].  Coastal topography further complicates redistribution patterns, and can also serve as
focal points for convection and associated convective downdrafts which redistribute and loft new dust in arid environments
(e.g., Grant and van den Heever 2014; Seigel and van den Heever 2012; Miller et al., 2008).
Such fine-scale features and interactions are inherently difficult to predict in numerical models, and require accurate
observations and coupled data assimilation techniques to produce and evaluate a representative analysis (Frolov et al., 2016;
Penny and Hamill, 2017; Zupanski, 2017).  Given the sparsity of applicable surface-based observations, particularly over
maritime regions, detection and characterization of littoral zone aerosol properties for short-term forecasting applications in
coastal zones worldwide is a problem best suited to satellite-based remote sensing.  Passive radiometer-based methods, the
most readily available form of satellite data, face their own array of challenges in coastal zones.  Shallow water and high-
turbidity conditions produce bright backgrounds which can obscure signal and produce biases in visible-based remote
sensing retrievals, and strong horizontal gradients in column moisture between continental and maritime air masses confuse
thermal infrared-based techniques.
Here, we consider a case where infrared-based detection performance varied significantly for ostensibly similar mineral dust
plumes adjacent to one another in the Southern Arabian Peninsula—one well detected and the other missed entirely. Through
a combination of remote sensing and modelling, we examine the impacts of the background environment on detection and
characterization of lofted dust in the Middle East. The epilogue of this 'tale of two dust storms' speaks to where, when, and
to what extent conventional infrared-based detection techniques are useful in various environments, and to the importance
consideration of the inherent uncertainties arising from natural variance in mineral dust characteristics.
This study is conducted as part of a Multi-disciplinary, University-led Research Initiative (MURI) conducted under the
auspices of the Office of Naval Research (ONR)—the Holistic Analysis of Aerosols in Littoral Environments (HAALE-
MURI).  A diverse team comprising expertise in numerical modeling, atmospheric aerosol physics, satellite-based passive
and active remote sensing, and data assimilation, has been assembled to gain a fundamental understanding of the principal
driving factors governing aerosol distribution, aerosol optical properties, and aerosol microphysical properties in coastal
regions.  The research includes development of new techniques for remote sensing and coupled data assimilation, toward
improved analysis and forecasting of parameters relevant to electro-optical propagation. It takes a holistic approach to this
challenge of understanding processes in a connected system as opposed to stand-alone analyses that may not fully account
for parameter coupling. The wide array of expertise brought to bear on this topic aims to build upon the community's
knowledge base and analysis tools for littoral-zone aerosol distribution, properties, and processes.



The paper is structured as follows. Section 2 takes inventory of existing satellite-based dust detection algorithms used
commonly for global aerosol mapping. Section 3 details from modelled and observed perspectives a case study showing a
widely varied performance of the popular "split-window" infrared-based dust detection technique. The sensitivity of the
split-window dust signal to water vapor and dust optical and geometric properties via idealized radiative transfer model
(RTM) simulations is examined in Sect. 4. Section 5 presents a new approach to the split-window technique based on
modulation of detection thresholds as a function of column water vapor information. The paper concludes in Sect. 6 with a
summary of the findings together with implications and recommendations for global satellite-based dust detection
algorithms.

## 9 2 Satellite-Based Dust Detection Methods

### 10 2.1 Basic Principles

The scientific literature is replete with satellite-based methodologies for lofted dust detection and characterization.
Ultraviolet techniques (e.g., Herman et al., 1997; Torres et al., 1998, 2007) take advantage of differences (spectral- and
angular-dependent) in backscattered radiances for absorbing and scattering aerosol species with respect to a molecular
atmosphere. Moving into the visible light and near-infrared wavelengths, algorithms take advantage of the preferential
absorption of blue light to enhance contrast over bright surfaces and vis-à-vis meteorological clouds (e.g., Miller et al., 2003;
Hsu et al., 2004, 2013; Qu et al., 2006). Combinations of visible/shortwave-infrared and thermal infrared are used to attain
contrast between dust over water and over land, respectively (e.g., Shenk and Curran, 1974; Ackerman, 1989; Tanré and
Legrand, 1991; Legrand et al., 2001; Cho et al., 2013). Recent advancements have enabled the use of O2 A and B
absorption bands and their nearby continuum channels to derive dust and smoke layer height over dark ocean and vegetated
land surfaces (Xu et al., 2017; Xu et al., 2018).

### 21 2.2 The Infrared Split-Window

The most commonly implemented techniques for satellite-based dust detection involve the Reststrahlen band of silica (or
quartz, a common and often significant constituent of mineral dusts found worldwide, Di Biagio et al., 2017), caused by the
Si-O bending mode in the 8-10 μm range (Peterson and Weinman, 1969; Salisbury et al., 1987; Wald et al., 1998). This
optical feature for mineral dusts results in elevated values of the real (scattering) and imaginary (absorption) parts of the
complex index of refraction for silicates, and commensurately higher values of extinction than surrounding wavelengths.
Passive radiometer narrowband channels positioned in the 8-12 μm 'atmospheric window' (a spectral region where the
gaseous atmosphere is largely transparent), used for imaging and describing the properties of meteorological clouds and the
land/ocean surface, can take advantage of the silica spectral fingerprint for dust detection. When used in tandem, the ~10



μm and ~12 μm narrowband spectral channels are often referred to as the 'split-window,' so-called because although this
part of the infrared spectrum is an atmospheric window, water vapor absorption is not entirely negligible, and is slightly
stronger at ~12 μm than at ~10 μm. Thus, this spectral band is 'split' between channels in relatively clean and more
absorbing portions of it. Hereafter, we will refer to the brightness temperature difference (BTD) between these two
wavelengths (i.e., T(10 μm) – T(12 μm)) as the split window BTD, or SWBTD.
Sensitivity of the SWBTD to atmospheric water vapor is well established, and it has long been incorporated as corrections to
retrievals of sea surface temperatures (e.g., McMillin, 1975), land surface temperature (e.g., Wan and Dozier, 1996), and
characterization of the lower atmosphere moisture itself (Lindsey et al., 2014). The enhanced absorption of water vapor at
12 μm produces a slightly positive BTD for conditions of a warm skin temperature and adiabatic lapse rate—a signal that is
of opposite sign to that produced by mineral dust. Thus, the dampening effect of water vapor on the SWBTD establishes a
working hypothesis for the missing dust plume of our Southern Arabian Peninsula case study.
Moving away from the reststrahlen band, the strength of dust extinction decreases gradually from 10 to 12 μm, resulting in a
negative SWBTD that can become significant (e.g., values between 0 and -6 K) when the underlying surface is warmer than
the lofted dust layer. Assuming other factors (e.g., dust layer height) are equal, the dust signal is most pronounced during the
daytime over deserts, when the skin temperature is much higher than the air temperature and the lapse rate of the lower
atmosphere is dry adiabatic. The optical properties of meteorological clouds differ from those of mineral dust, making the
SWBTD test useful for isolating lofted dust layers in a cloudy scene. Optically thin cirrus clouds typically exhibit a
relatively large positive SWBTD—opposite to the dust signal. Optically thick clouds tend to show near zero to slightly
positive values of the SWBTD. However, intervening clouds (above the dust layer) will obscure and mask-out the SWBTD
dust signal, limiting the applicability of this passive detection satellite technique to cases of cloud-free line-of-sight from the
satellite to the dust layer.

## 2.3 Satellite Techniques Enlisting the Split Window

The European Organisation for the Exploitation of Meteorological Satellites (EUMETSAT; www.eumetsat.int) demonstrates
a Red/Green/Blue (RGB) dust enhancement technique (EUMETSAT Dust RGB; e.g., Lensky and Rosenfeld, 2008) that
takes advantage of the SWBTD dust signature. The appeal of the infrared-based techniques is their 24-hr utility, although
the strong dependencies on the atmospheric profile and the diurnal pattern of surface temperature can make for varying
performance and ambiguity. Over deserts, the surface possesses a similar mineralogy and radiometric behaviour to the
locally lofted dust, producing SWBTD false-alarms. Over ocean surfaces, where the temperature of the water may be close
to or cooler than the lofted dust layer, the SWBTD dust signal is inherently weaker, and does not become significant unless
the dust layer itself is optically thick. These challenges are readily apparent in the EUMETSAT RGBs, but are overcome in





part by the application of these algorithms to geostationary data, where motion helps analysts to differentiate between actual
dust and false-alarms locked to the surface structures.
Work-arounds to the surface signal ambiguity problem have been attempted (e.g., Legrand et al., 2001; Tramutoli et al.,
2005, 2007).  Miller et al. (2017) employ a front-end cloud mask with a priori information on the clear-sky background
surface emissivity to identify and suppress the undesired enhancement of land surfaces, via a Dynamic Enhancement
Background Reduction Algorithm (DEBRA). However, DEBRA's ability to detect lofted dust faces the same physical
limitations as the EUMETSAT Dust RGB Product, as it enlists the same spectral bands (and associated physics) to identify
dust layers.
In both SWBTD-based dust detection techniques, effects of water vapor absorption in the lower atmosphere will impact
performance. Although dust characteristically is associated with arid environments and inherently lower column water vapor,
cases do arise when dust is lofted within or transported into anomalously moist (with respect to regional climatology) air
mass environments. The effects of SWBTD dust signal suppression by water vapor in terms of moisture amount, dust
amount, temperature profile, and positioning of the dust layer within the moist profile, have not been examined
systematically in the literature.  We begin to explore some of these questions here.
**3 Case Study Description:  Middle East Dust Storms, 3-4 August 2016**
A highly useful case allowing us to explore the impacts of water vapor on SWBTD occurred during early August 2016,
when two dust storms met in east/west alignment over the Southern Arabian Peninsula of Southwest Asia.  The western dust
storm originated on the morning of 3 August over the Saudi Arabian interior, near its northeastern borders with Iraq and
Kuwait, and moved south/southeast.  The eastern storm formed later that day and evening along the southeastern portion of
the Arabian Peninsula and moved northward.  On the morning of 4 August and over the next two days, these two large dust
plumes juxtaposed but did not mix—remaining bound to the air masses that carried them.



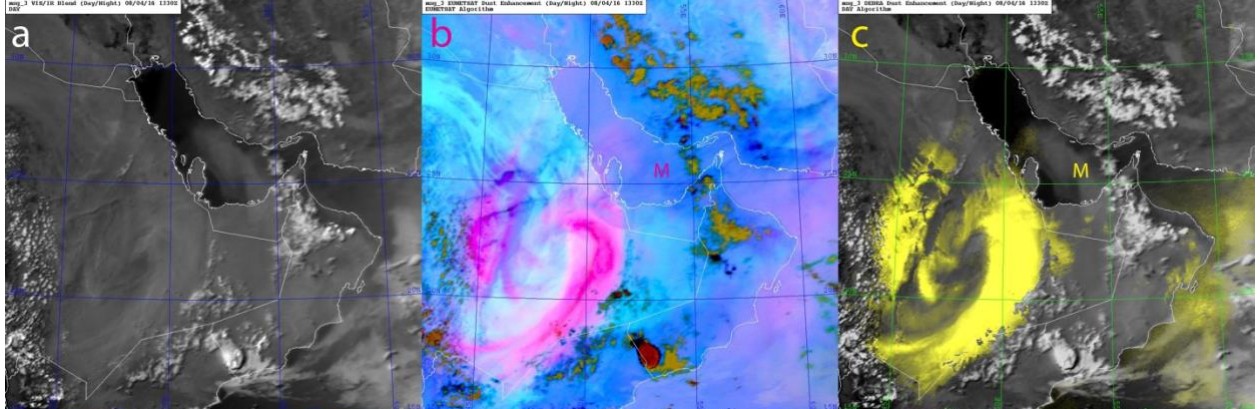

**Figure 1: Meteosat Second Generation-8 satellite imagery on 4 August 2016 at 1300, for a) visible reflectance, and infrared-based**
**significant dust enhancements from b) EUMETSAT and c) DEBRA algorithms. Labels "M" in panels (b) and (c) denote the over-**
**water portion of a significant dust plume that was missed almost entirely by the infrared-based dust detection methods. Here, the**
**missing dust plume is only evident in visible imagery (panel a) due to its high contrast against the dark ocean background.**
Figure 1, showing Meteosat Second Generation-8 (MSG-8) imagery collected on 4 August at 1300 UTC (about 4 PM local
time in Qatar, located in the center of these frames), illustrates our motivation for studying these two disparate dust storms.
The left panel shows visible reflectance imagery, while the center and right panels show algorithmic enhancements of dust
via the EUMETSAT Dust RGB Product and the DEBRA technique of Miller et al. (2017), respectively. Noteworthy is the
markedly different performance of the SWBTD-based dust enhancements for the two dust plumes. Both methods readily
capture the northern-originating inland (western) dust plume but completely miss the southern-originating (eastern) plume
entering the Southern Persian/Arabian Gulf (hereafter, SG).
Associated with these two storms were marked differences in air mass properties—the western storm was embedded within a
dry continental air mass (low column-integrated moisture as measured by Total Precipitable Water, TPW), and the eastern
storm within a moist maritime air mass originating from the northern Arabian Sea. It is hypothesized that the dry and moist
air masses associated with these dust storms played a governing role in the varied performance of SWBTD-based satellite
dust detection algorithms. To better understand this possible linkage, we begin with a discussion of the meteorological
conditions, surface/satellite observations, and numerical model analysis characterizing this case.
**3.1 Meteorological Set-Up: Synoptic and Mesoscale Forcing**
The following is a synopsis of the case study based on analysis of numerical modeling and surface station observations. The
Navy Global Environmental Model (NAVGEM; Hogan et al. 2014) and the Weather Research and Forecasting model
coupled to Chemistry (WRF-Chem; Grell et al., 2005) model were used to interpret aspects of the synoptic and mesoscale
conditions for this case. WRF-Chem was coupled to a Naval Research Laboratory (NRL) dust source database for Southwest



Asia (Walker et al., 2009) to examine dust lofting and transport. Figure 2 shows a NAVGEM analysis of the 500 mb
geopotential height field and the relative vorticity during the initial stages of the event. Figure 3 shows WRF-Chem
simulations for the column integrated dust mass for selected times across the 3-4 August period leading up to the satellite
imagery shown in Fig. 1. Both dust plumes seen in satellite imagery were captured by this coupled model system. This case
study was also simulated using the Regional Atmospheric Modeling System (RAMS; Cotton et al 2003; Saleeby and van den
Heever 2013), and differences in the amounts of dust lofted between the WRF and RAMS simulations are detailed by
Saleeby et al. 2018.

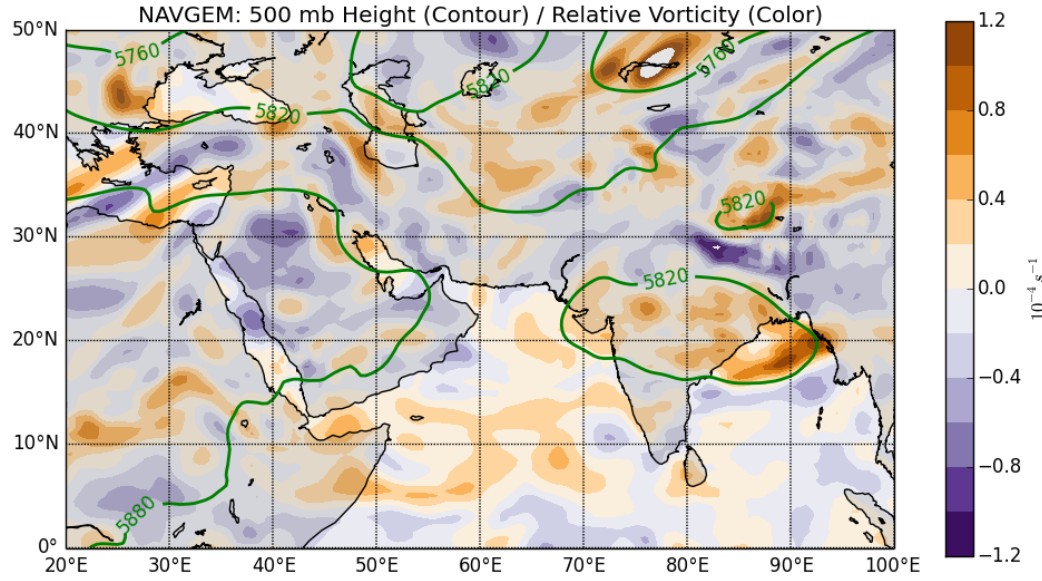

**Figure 2: NAVGEM 500 mb geopotential height and relative vorticity analysis for 00Z August 3, 2016.**





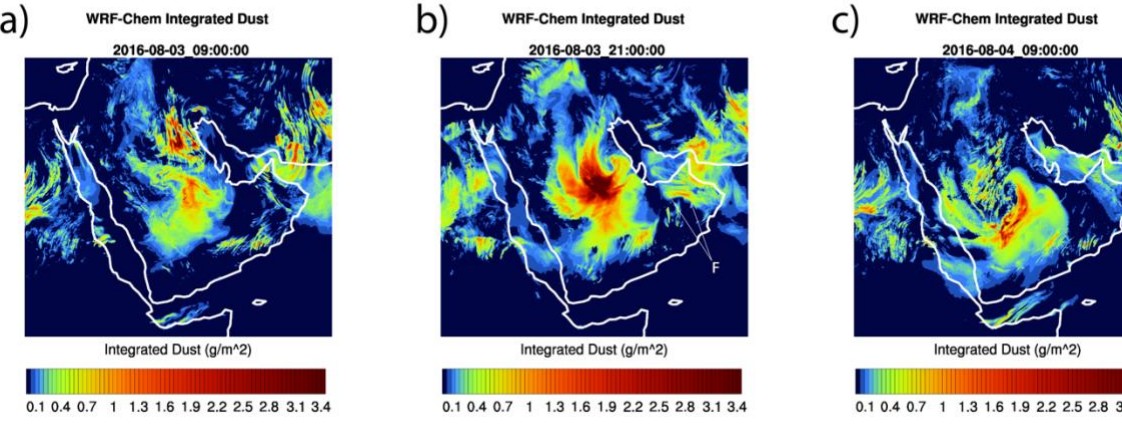

**Figure 3: WRF-Chem simulated vertically integrated dust (g/m$^2$) on: a) 3 August 2016, 0900 UTC, b) 2100 UTC, and c) 4 August at 0900 UTC leading up to the 4 August observations of missing dust shown in Fig. 1. The feature "F" in panel (b) denotes pooling of dust along a southerly surge of air near the surface, responding to the low-pressure system in the SG.**

On 2 August 2016 at 00Z, NAVGEM analysis (Fig. 2) shows a broad 588 dm geopotential height upper-level ridge over northeast Africa and the Arabian Peninsula. On 00Z of 3 August, a southward moving shortwave over southern Iraq started to traverse around this high-pressure ridge and into the central Saudi Peninsula. Between 05-06Z on 3 August, this shortwave intensified the winds from 700 mb down to the surface, mobilizing dust from southern Iraq across the Ad Dahna desert to the Rub Al Khali over the eastern edge of the Saudi plateau. This dust was transported within a very dry air mass (TPW < 20 mm; discussed in Sect. 3.4). As the shortwave continued to move around the ridge and southwestward on 4 August, this lofted dust was readily observable over the Saudi plateau in various dust-enhanced satellite imagery products.

Between 12Z and 18Z on 3 August, both model and surface observations showed a mesoscale low-pressure system forming over the SG east of Qatar. The alignment of the Al Hajar mountains in northern Oman and the Zagros mountains in southwestern Iran aided in the blocking and redirection of the surface winds, reinforcing the cyclonic flow around the surface low in the SG. Strong southerly winds formed in response to this surface low, mobilizing dust from 3 August 18Z to 4 August 09Z over an area extending from Oman to the coast of the United Arab Emirates (UAE).

Over this period, surface stations from the Oman coast to the UAE coast reported southerly winds at 5-20 kts (2.6 -10.3 m/s), accompanied by reports of dust in suspension, dust raised by winds at the time of observation, dust devils, and slight-to-moderate dust storms. Evidence of dust lofting and pooling along a surface front formed by this southerly surge, as captured



by WRF-Chem coupled to the NRL dust source database, is noted in Fig. 3b (animation provided as Supplemental
Information S1). Convection initiated in this region may have produced haboobs (Miller et al., 2008) that contributed as a
secondary source to the model's total dust loading. The correct placement and timing of such convection is a major challenge
of numerical modeling, and thus poses a source of uncertainty in regional dust forecasting. Dust from these various sources
comprised the eastern plume, which was transported into the SG on 4 August (Fig. 1).
Surface observations collected across the southeastern portion of the Arabian Peninsula over 09Z-12Z on 4 August show that
the surface winds weakened across the region over this period, although dust storms and blowing dust conditions prevailed.
By 18Z, nearly all stations across the region reported dust in suspension, with only one station still reporting an active (i.e.,
blowing) dust storm. Although the surface dust lofting event had ended by this time, satellite imagery showed copious dust
aloft, being transported northward. This dust remained in suspension over the region for the following two days before
being dispersed by the synoptic-scale flow or settling out.

### 3.2 Satellite Observations: Passive Sensors

Figure 4 shows true color and DEBRA dust-enhanced imagery for a zoomed-in portion of Fig. 1 as observed by the
Visible/Infrared Imaging Radiometer Suite (VIIRS) instrument on the Suomi National Polar-orbiting Partnership (Suomi-
NPP) satellite. In Fig. 4a, the tan coloration of the plume in the SG (as noted by label "M" in Fig. 1 b and c) is characteristic
of lofted dust. Although not as obvious over the bright desert interior due to poor brightness contrast, low color contrast, and
relatively low surface variability, the true color imagery shows this plume originates from inland portions of eastern UAE,
flanking the Al Hajar mountain range in northern Oman. The preferential absorption of blue-wavelength light by silicates,
responsible for the perceived tan coloration, was keyed on by visible-based dark-target dust enhancements from the National
Aeronautics and Space Administration (NASA) Aqua satellite's Moderate-Resolution Imaging Spectroradiometer (MODIS)
(e.g., Miller, 2003; not shown) as well, confirming the composition of the plume as silicate dust, as opposed to
meteorological clouds, biomass smoke, or pollution. This mineralogy would also be expected to produce a distinct negative
SWBTD (dust) signal, and thus be readily enhanced by DEBRA, but this enhancement was not achieved in this case.
Instead, DEBRA missed this feature (i.e., not enhanced as yellow in Fig. 4b) almost entirely.
DEBRA includes a cloud mask preprocessing step, which enlists the SWBTD as a 'restoral test' for pixels erroneously
flagged as cloud instead of optically thick dust. It was confirmed, by temporarily turning it off in the processing, that the
cloud mask itself was not the root cause for the missing dust in the original DEBRA algorithm. As the EUMETSAT Dust
RGB algorithm (Fig. 1b), which also misses the eastern dust entirely, does not enlist a cloud mask, this exercise was done
simply as a sanity check before pursuing alternative explanations for the missing dust.



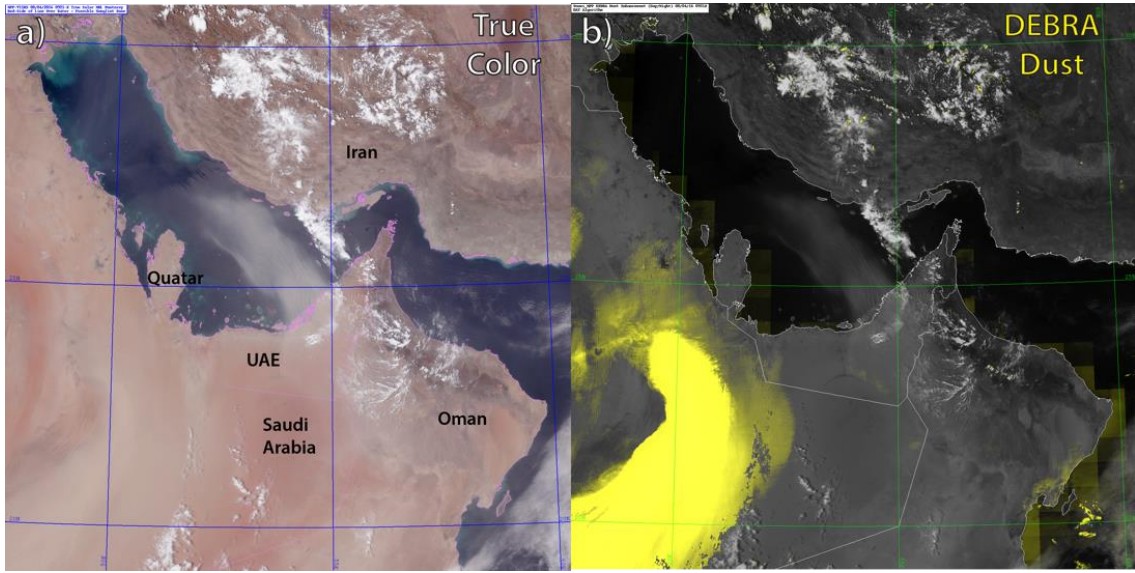

**Figure 4: Suomi-NPP VIIRS true color (a) and DEBRA-dust (b; enhanced dust in yellow) imagery composite from 4 August 2016, 0921 UTC. Whereas DEBRA identifies the western dust plume prominently, it fails to identify the eastern 'missing' plume over the eastern UAE, extending offshore. The tan coloration of this plume is revealed via true color imagery, in contrast to nearby meteorological clouds (white).**

Geostationary MSG-8 visible-band imagery (time-resolved at 30 min, provided as Supplemental Information S2) provides insight into the additional factors contributing to the eastern dust plume. The imagery shows morning-time convection initiating offshore of Oman, between the coastal cities of Lakabi (18.23N, 56.55E) and Ras Madrakah (18.98N, 57.79E), on 3 August 2016 at 0400 UTC (~8 AM local time). By 0815 UTC, a well-defined boundary marked by leading-edge cumulus formed and began moving inland in a northwesterly direction. Such features are indicators of a cold pool outflow and gust front associated with evaporative cooling, common to convection occurring in dry lower atmospheric environments. This outflow may have produced additional dust lofting via the haboob mechanism (e.g., Miller et al., 2009). The gust front appears to superimpose with the daytime inland-moving sea breeze front along coastal Oman. The presence of dust was inferred by high visible reflectance and reduced land surface texture behind the front, although standard dust-enhancement products (SWBTD) gave no indication of lofted dust.

By 1300 UTC (5 PM local time) on 3 August 2016, the coastal Oman dust-laden front had just crossed the border between Oman and southern Saudi Arabia at 20N, 55E. Infrared signatures of clouds oriented along the front were lost by 1800 UTC. By this time, the front had come under the influence of southerly flow (and principal source of dust lofting in the missing plume, described in Sect. 3.1), taking a more northward track along the Saudi/Omani border, moving toward the southern border of the UAE. Both visible (Fig. 1a) and true color (Fig. 4a) imagery from the following day show a dust plume which



consists of both local and secondary regionally-lofted sources as described, extending through central UAE and nosing
offshore into the SG.
The well-detected and missed dust plumes as observed in juxtaposition on 4 August 2016 contained very different spectral
properties across the infrared window where the SWBTD operates. Figure 5 shows information from Aqua MODIS and
Atmospheric Infrared Sounder (AIRS) instruments. Fig. 5a shows the domain of interest via MODIS true color imagery and
denotes locations of selected AIRS spectral plots. Locations 1-2 correspond to offshore and onshore locations within the
SWBTD-missed dust plume, and Location 3 is within part of the SWBTD-detected dust plume. Fig. 5b shows MODIS
retrievals of aerosol optical depth (AOD) via the visible-based "Deep Blue" algorithm. Fig. 5c shows a field of SWBTD
from AIRS, where negative values shown in blue correspond to dust detections.





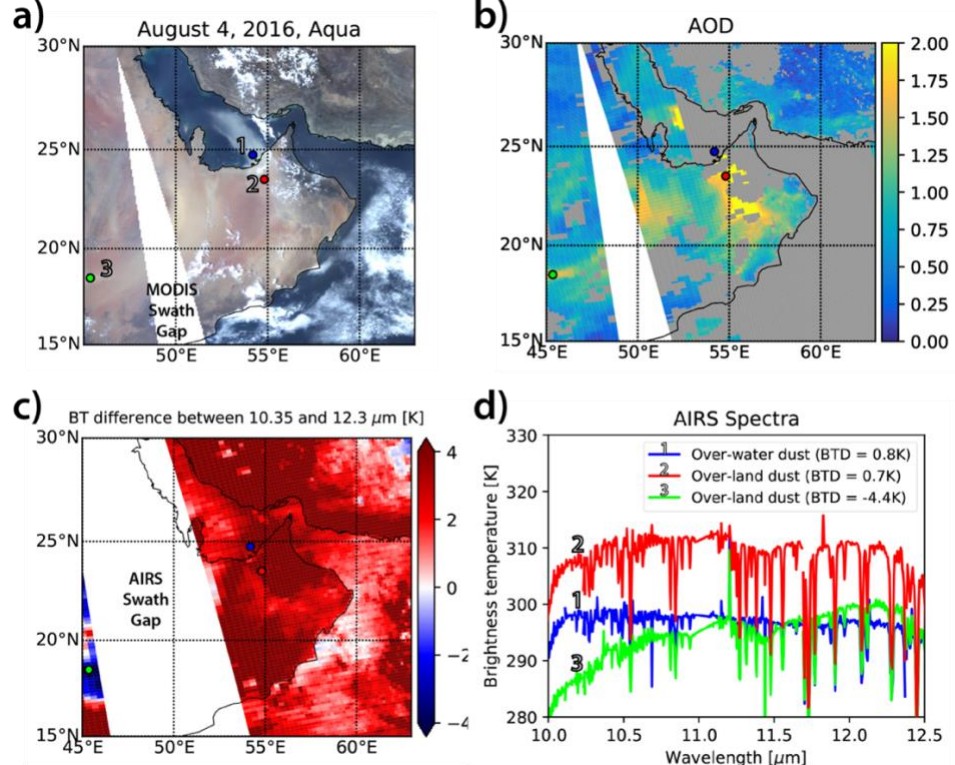

**Figure 5: (a) MODIS true color imagery for 4 August 2016, with sample locations for the missed dust plume over-water (1) and over-land (2), and a southwestern portion of the extensive and well-detected inland plume (3). (b) Corresponding MODIS AOD retrieval. (c) AIRS-derived SWBTD at locations noted in (a). (d) AIRS split-window region spectra for locations noted in (a), with SWBTD values labeled.**

AIRS spectra (and associated SWBTD values) at the selected dust plume locations noted in Fig. 5a are shown in Fig. 5d. The 1650 km wide swath width of AIRS (narrower than the 2330 km swath of MODIS) accounts for the differences in the width of data gaps between adjacent satellite overpasses. While the AIRS swath gap on this day corresponded to a significant portion of the well-detected dust plume, a portion of the western swath intersects the southwestern edge of the same storm (near Location 3), and thus is relevant to the comparison of dust signals between the two air masses. Of note in Fig. 5d is that, whereas Location 3 produces a strongly negative SWBTD, indicative of lofted dust, Locations 1 and 2 produce small/positive SWBTD values—explaining the inability of the EUMETSAT and DEBRA methods to infer the presence of dust at those locations.

### 3.3 Vertical Profile Information from Passive Sensors

Passive imaging radiometers, while providing useful information on the horizontal distribution and column-integrated properties of dust storms, offer very limited information in terms of vertical structure. Active optical sensors such as lidar



are very useful in this regard, but terrestrial systems are few and far between.  Even less common are spaceborne lidars,
which extend coverage to the global scale but typically provide only a curtain slice of the column owing to their non-
scanning configuration.  This trait, combined with their operation on polar-orbiting satellites offering only infrequent revisits
to any specific location, renders the chances of leveraging such observations opportunistically for a localized case study very
small.   At the time of this case study, two NASA cloud/aerosol lidar systems were operational—the Cloud-Aerosol Lidar
with Orthogonal Polarization (CALIOP; Hunt et al., 2009) on the NASA Cloud-Aerosol Lidar and Infrared Pathfinder
Satellite Observation (CALIPSO; Winker et al., 2009) and the NASA Cloud-Aerosol Transport System (CATS; McGill et
al., 2015) on the International Space Station (ISS).
By what can perhaps best be described as a stroke of incredible serendipity, both the ISS and CALIPSO satellites carrying
these lidar systems crossed over the same region of the UAE within minutes of each other (nearly a simultaneous nadir
overpass) on the evening between 3-4 August 2016.  Moreover, the conjunction occurred at the very time that the 'missing'
dust plume was forming and being transported northward by various mechanisms described in Sect. 3.1 and 3.2.  Traveling
from NW to SE on the ISS, CATS crossed over the UAE on 3 Aug at ~2208 UTC.  Roughly 15 min later, at ~2223 UTC,
CALIOP crossed the same area on its NE to SW descending node track.  This conjunction of active systems provided a
golden opportunity to assess the vertical structure of the missing dust plume at a critical time in its development.



Figures 6 and 7 show the satellite ground tracks and corresponding lidar profiles of total attenuated backscatter for CATS
and CALIPSO, respectively. Differences seen in the magnitudes of total attenuated backscatter are due primarily to the
different wavelengths (1064 nm for CATS, and 532 nm for CALIOP) shown in these figures. Both profiles indicate a deep
layer (0-5 km above mean sea level, AMSL) of suspended aerosol (identified by CALIPSO algorithms, not shown, as likely
dust), common to the region during this time of year (e.g., Nabavi et al., 2016). Embedded within this background are more
significant backscatter features tied to local, recently lofted and optically thicker dust. Meteorological clouds, especially
those in the middle/lower atmosphere composed of mixed or liquid phase droplets, act as very strong attenuators of lidar
energy. Attenuation from these clouds, also present within the CATS and CALIPSO profiles, accounts for the occasionally-
seen vertically-oriented dark stripes/bands in the lower atmosphere of both lidar profiles.



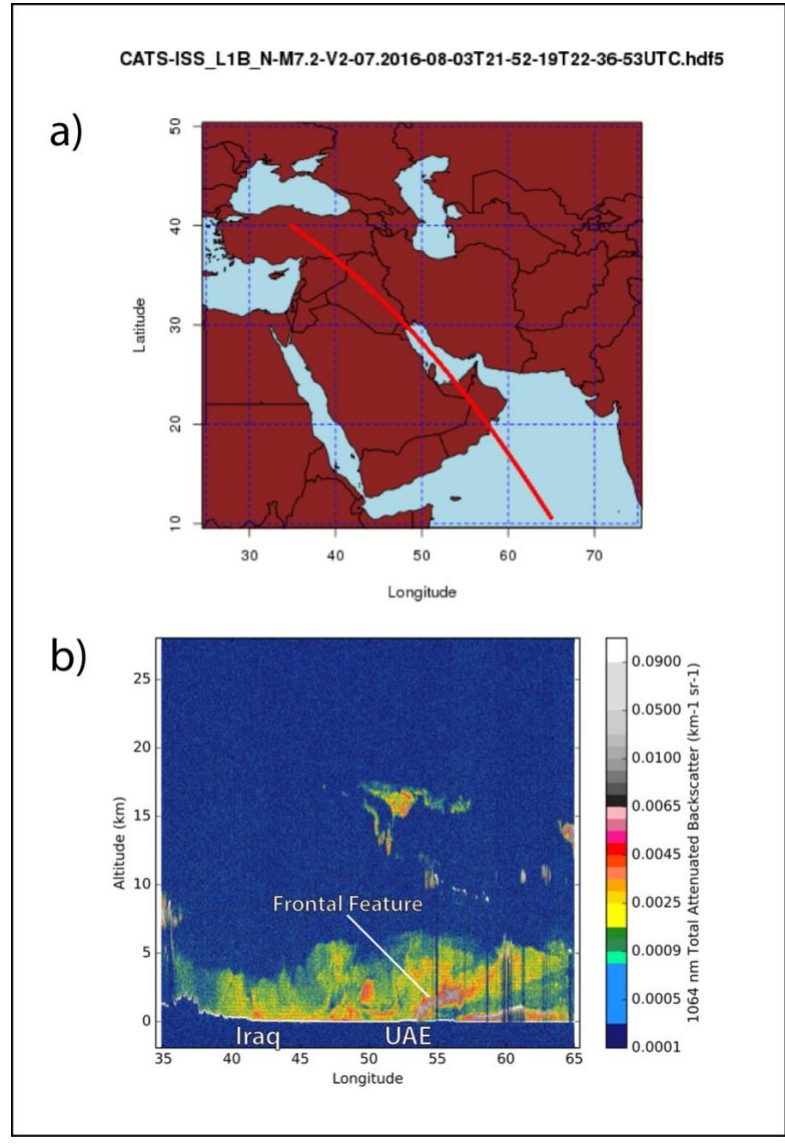

2 **Figure 6: (a) ISS ground track and (b) CATS 1064 nm total attenuated backscatter cross section. Noted is an apparent dust**

3 **frontal feature extending from the surface to 2-3 km above mean sea level.**

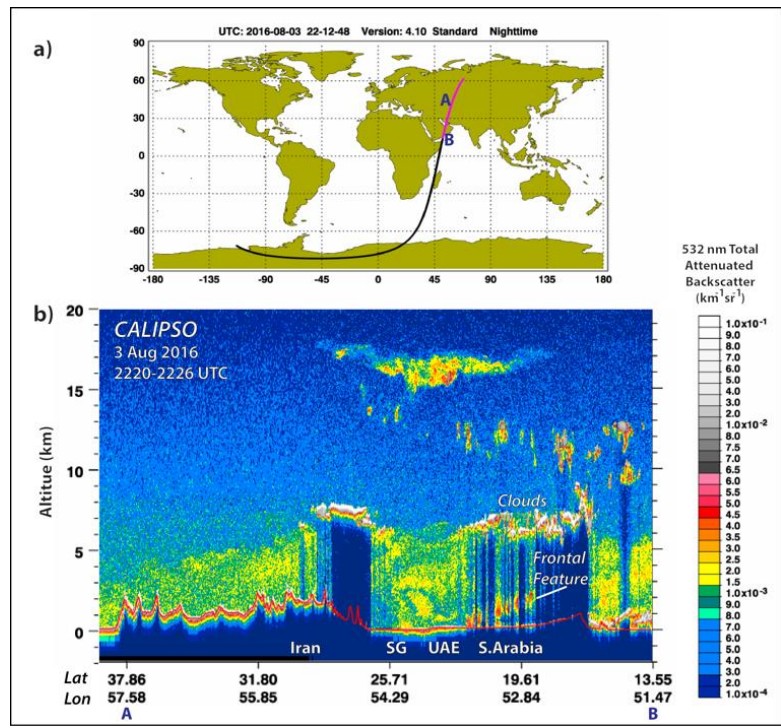

**Figure 7: (a) CALIPSO ground track and (b) CALIOP 532 nm total attenuated backscatter cross section (for the subset of this**
**track from locations A to B as noted). The same dust frontal feature noted in Fig. 6 is present near the southern UAE border with**
**Saudi Arabia.**
A key item of interest in these lidar profiles is the labeled '*Frontal Feature*,' seen in both CATS and CALIPSO observations
over the southeastern UAE. This feature arcs from the surface up to ~2-3 km. The diffuse character of its lower and upper
boundaries, combined with its significant backscatter (entering the range of the color scaling typically relegated to cloud
signals), are signs of an optically thick dust layer. In the CATS data of Fig. 7, the same arc feature seen in the CALIPSO
data of Fig. 6 is present but appears less continuous, due to the intermittent beam attenuation by the overriding mid-level
clouds encountered by the CALIPSO ground track but avoided by the W/NW approach of the ISS.
The timing and location of the frontal feature are consistent with the WRF-Chem and RAMS model simulations and surface
station observations of a southeasterly flow surge associated with the forming low pressure over the SG (Sect. 3.1). The
southbound components of both CATS and CALIPSO would have encountered the leading edge of this frontal structure.
The effects of such a surge on freshly lofted dust and dust already in suspension would be for a pooling and riding up and
over the surface front, consistent with the lidar observations. Both effects are identified in the model simulations: in Fig. 3b,
the area labeled "F" shows the bowing of suspended dust along the southerly surge, as well as northbound streamline plumes





characteristic of freshly lofted dust forming in its wake. From these dual lidar observations, we infer that the 'missing' dust
plume has significant mass loading in the 0-3 km layer (~1000 to 700 mb) of the atmospheric column.
**3.4 Atmospheric Moisture**
We turn our attention now to the topic of atmospheric moisture, which is hypothesized to have played a governing role in the
performance (or lack thereof) of the infrared-based dust detection techniques. Figure 8 shows WRF-Chem simulations,
corresponding to Fig. 3, of the 24 hr period leading up to the 4 August 2016 event depicted in Fig. 1 (animation provided as
Supplemental Information S3) The simulations show a very dry (low-TPW) air mass in cyclonic rotation (the shortwave
discussed in Sect. 3.1) descending through central Saudi Arabia, associated with the well-detected dust plume. By
inspection with Figs. 1, 4 and 5, this dry air mass correlates with the well-detected western dust plume. Meanwhile, a tongue
of high-TPW extends across Oman and northward through the eastern UAE. Strong southerly winds entrained relatively
moist air (TPW > 45 mm) from the Arabian Sea maritime air mass. The frontal surge that occurred over the evening hours
of 3-4 August is manifest in Fig. 8b (compare to Fig. 3b) as an enhanced gradient of moisture. Cross-referencing once again
to observations in Figs. 1, 4, and 5, this moist air mass envelopes the region of the 'missing' eastern dust plume.

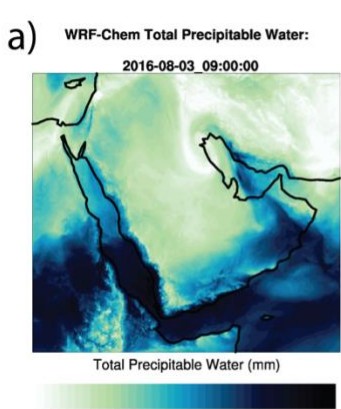 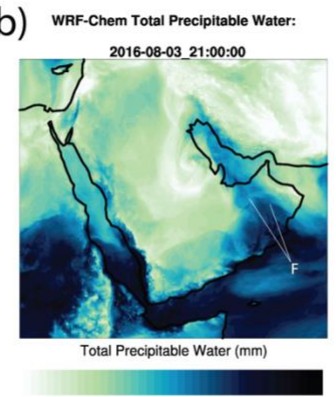 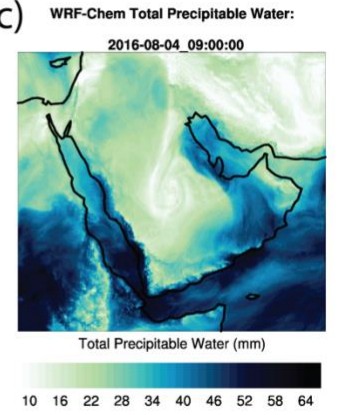

**Figure 8: As in Fig. 3., but showing Total Precipitable Water (TPW; mm). The frontal feature denoted in Fig. 3b and discussed in**
**Sect. 3.3 is shown again here as "F" in panel (b).**
Properties of the dry and moist atmospheric air masses were also characterized directly by independent satellite retrievals.
The National Oceanic and Atmospheric Administration (NOAA) Unique Combined Atmospheric Processing System
(NUCAPS; Gambacorta and Barnet, 2013), run operationally since 2013, uses cloud-cleared radiances and an iterative
regularized least squares minimization algorithm to produce vertical profiles of temperature, water vapor, and trace gases




from microwave and infrared radiances. The retrieval enlists input from the Cross-track Infrared Sounder (CrIS) and the
Advanced Technology Microwave Sounder (ATMS) on NOAA's Suomi-NPP and Joint Polar Satellite System-1 (JPSS-1)
satellites. NUCAPS provides thirty retrievals at 100 levels between 1100 and 0.16 hPA, across a 2200 km swath. Owing to
the projection of the cross-track scanning sensor footprints, spatial resolution ranges from ~50 km at nadir to ~70 x 134 km
at scan edge. Nalli et al (2016) demonstrate that NUCAPS water vapor profiles compare favorably to radiosonde data and to
numerical weather prediction models in terms of moisture magnitude and gradients in a variety of atmospheric flows.

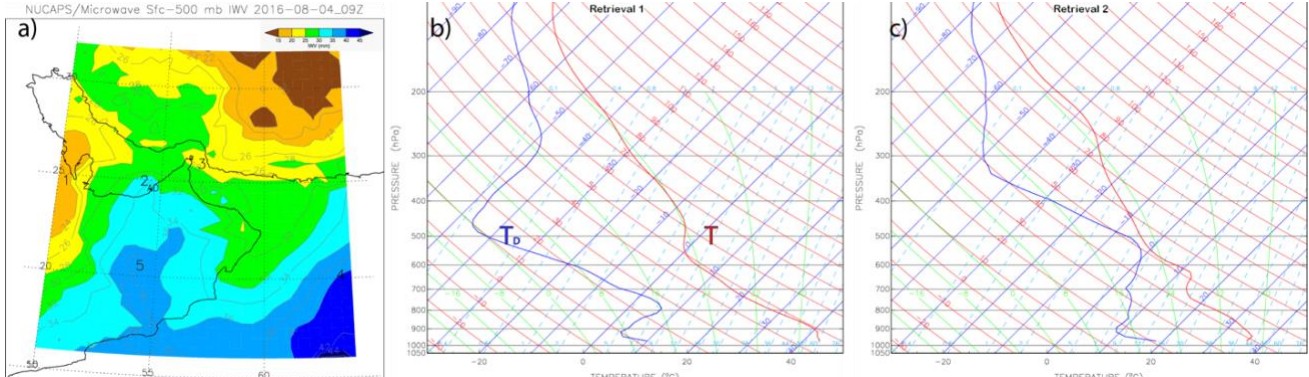

**Figure 9: NUCAPS derived integrated water vapor between the surface and 500 mb (a) valid on 4 August 2016 at 0921 UTC, and**
**profiles of temperature (T) and dew point temperature (T$_D$) at two selected locations: b) the dry air mass (location 1, west of**
**Qatar), and c) moist air mass (location 2, offshore of UAE in the SG). Note the significant differences in low- to mid-level moisture**
**between the two soundings.**
Figure 9 shows NUCAPS retrievals for the Suomi-NPP pass over Southern Arabian Peninsula on 4 August 2016 at 0921
UTC (same overpass as shown in Fig. 4). A plan view of low- to mid-troposphere (surface to 500 mb) Integrated Water
Vapor (IWV; Fig. 9a), also referred to as Total Precipitable Water (TPW), shows a tongue of moisture extending from
coastal Oman to the SG. The structure of this moist air mass, drawn northward by the surface low over the SG as discussed
in Sect. 3.1, is consistent with the WRF-Chem analysis shown in Fig. 8. Selected NUCAPS temperature and dew point
temperature profiles for two locations within portions of the dry and moist air mass dust plumes are shown in Figs. 9b and
9c. The NUCAPS profiles show significant differences in low/mid-tropospheric moisture, especially at mid-tropospheric
levels around 500 mb.
These diverse model and observational datasets provide a comprehensive understanding of the meteorological conditions
associated with the case study. Further, they motivate a more in-depth analysis, based on RTM simulations and in
consideration of varying dust optical properties, of the potential impacts of atmospheric moisture differences on the SWBTD
dust signal. This analysis is aimed at determining the necessary conditions for producing the 'missing' dust outcome in the



case study, while looking toward comprehensive methods of mitigating the water vapor masking effect on infrared detection
techniques in general.
**4 Idealized Dust Sensitivity Analysis**
**4.1 Microphysical and Optical Property Assumptions**
Differences in SWBTD among various dust plumes may arise from water vapor masking, different optical properties, optical
depth, or more likely, a combination of all factors. The composition of mineral dust varies as a function of location, yielding
different optical properties (and commensurate SWBTD signal strengths) for a given mass loading and size distribution.
Thus, for the current case study it is important to take inventory of variations due a range of refractive indices (RI)
characteristic of dust sources over the Arabian Peninsula, and consider to what extent these variations might account for the
disparate behaviours of the IR dust detection performance in the two dust plumes.
Several sets of RI for varying dust types characteristic of the Arabian Peninsula reside in the literature. For this analysis, we
enlisted recommendations from i) the quartz-dominated (99% quartz, 1% hematite) species from the Aerosol Refractive
Index Archive (ARIA; http://www.atm.ox.ac.uk/project/RI/minerals.html), ii) the Optical Properties of Aerosols and Clouds
database (OPAC; Hess, et al., 1998), and iii) Saudi Arabian dust properties from Di Biagio et al. (2017; hereafter DB17).
The values of these RI databases, along with spectral optical properties derived from Mie theory for 2.4 μm effective radii
dust at wavelengths used in constructing the SWBTD, are provided in Table 1. Mass extinction coefficients ($k_{ext}$; m²/kg) are
multiplied by the dust loading (mass mixing ratio), the density of air, and the geometric thickness of the dust-laden model
layers, and the optical thicknesses of each layer are then combined to yield the total AOD. The single scatter albedo
describes the fraction of the extinction that is due to scattering processes (vs. absorption), and the asymmetry parameter
provides a metric for directional scattering (1.0 denotes complete forward scatter, -1.0 denotes complete backscatter, and 0.0
denotes isotropic scatter).

| Wavelength (μm) | ARIA (RI, $k_{ext}$, $\omega_o$, g) | | | | OPAC (RI, $k_{ext}$, $\omega_o$, g) | | | | DB17 (RI, $k_{ext}$, $\omega_o$, g) | | | |
|---|---|---|---|---|---|---|---|---|---|---|---|---|
| 10.35 | 2.39 - 0.036i | 340.3 | 0.871 | 0.375 | 2.50 - 0.5i | 318.1 | 0.495 | 0.469 | 1.620 - 0.115i | 174.4 | 0.642 | 0.605 |
| 12.30 | 1.49 - 0.06i | 89.5 | 0.692 | 0.568 | 1.65 - 0.5i | 201.5 | 0.347 | 0.505 | 1.508 - 0.018i | 83.6 | 0.889 | 0.564 |

**Table 1: Refractive indices (RI), mass extinction coefficient ($k_{ext}$), single scatter albedo ($\omega_o$) and asymmetry parameter (g) assumed**
**for idealized SWBTD dust signal calculations.**



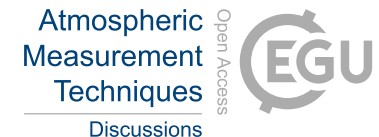

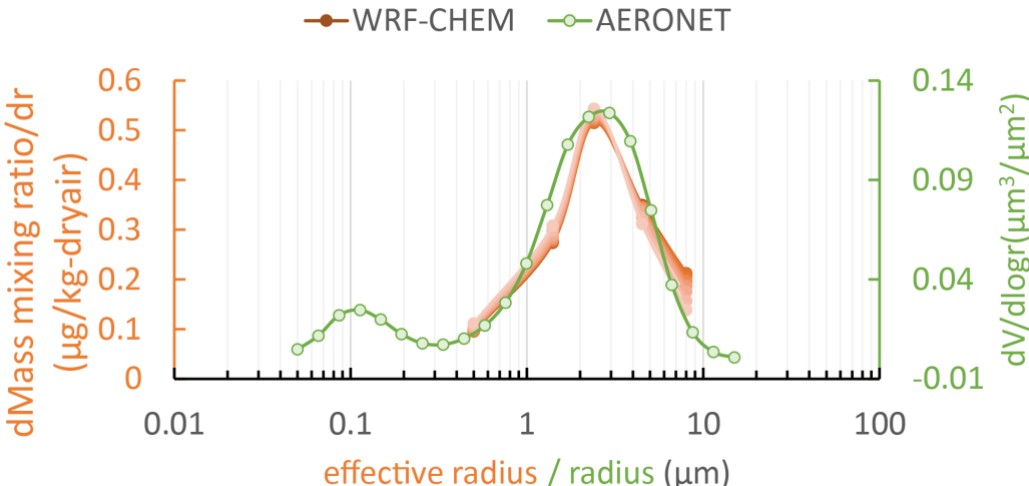

**Figure 10: Comparison between the volume size distribution of AERONET data (2016 average) at Kuwait University site and the mass distribution of dust-laden columns in WRF-Chem.**

The spherical particle approximation of Mie theory was assumed in computing the dust optical properties. A mean dust radius of 2.4 μm was used in the idealized RTM calculations, corresponding to the large particle radius mode of an averaged dataset collected at the Kuwait University AERONET site during 2016 (Figure 10). Over-plotted on Fig. 10 is the averaged WRF-Chem model binned size distributions for atmospheric columns having total summed dust volume concentrations (dust loading) greater than 0.6 μm$^3$/μm$^2$ for the current case study, showing that the model assumptions were consistent with the AERONET-observed large particle radius mode. This representativeness is important for simulations in Sect. 4.3.

| | AOD(10.35 μm) | | | AOD(12.30 μm) | | |
|---|---|---|---|---|---|---|
| Dust Loading | ARIA | OPAC | DB17 | ARIA | OPAC | DB17 |
| 40 μg$_{dust}$ / kg$_{air}$ | 0.253 | 0.237 | 0.130 | 0.067 | 0.150 | 0.062 |
| 80 | 0.507 | 0.474 | 0.260 | 0.133 | 0.300 | 0.208 |
| 186 | 1.183 | 1.106 | 0.606 | 0.311 | 0.701 | 0.291 |

**Table 2: Examples of AOD for 10.35 μm and 12.30 μm for three different dust loading amounts, using dust RI and optical property assumptions from Table 1 and an assumed dust layer thickness of 2 km.**

Table 2 shows example values of AOD computed at the SWBTD component wavelengths for three different total column dust loadings (40, 80, 186 in units of columnar volume concentration; μg$_{dust}$/kg$_{air}$) and the three sets assumed of RI (Table 1), based on Mie theory for dust effective radius of 2.4 μm. At 10.35 μm, the real part of the RI (which dominates the extinction via scattering) is much lower for DB17 than the OPAC and ARIA databases, resulting in an AOD that is about a





factor of 2 smaller. At 12.3 µm, the ARIA and DB17 RIs produce similar AODs, while OPAC is significantly larger—the
difference in this case being OPAC's larger imaginary RI component (translating to enhanced absorption). Despite these
differences, the proportional relationship of extinction between the two wavelengths does not change among the various
composition assumptions. Hence, the sign of the SWBTD for mineral dust remains negative for all three databases.
However, the magnitude of the SWBTD for a given RI database will vary as a function of different
temperature/moisture/dust profile scenarios.

## 4.2 Radiative Transfer Calculations

Corresponding values of SWBTD from the three dust RI databases were calculated using the hybrid Eddington RTM of
Deeter and Evans (1998), following its implementation by Grasso and Greenwald (2004). Figure 11 shows results for
idealized single dust layers having optical properties corresponding to ARIA, a mean particle size of 2.4 µm, residing in an
adiabatic temperature profile with surface temperature of 300 K. A 2-km thick dust layer of variable aerosol optical depths
(AOD, referenced at 10.35 µm) shown (from clear sky to optically thick, AOD > 10) was raised through the atmospheric
column, and the column moisture was increased linearly, accounting for the change in TPW along the x-axis. The AOD of
(0.254, 0.507, 1.183) are a function of dust loading values of (40, 80, 186; µg(dust)/kg(dry air)), respectively. The y-axis of
Fig. 11 shows magnitude of the computed SWBTD. Included is the clear-sky reference (AOD = 0), which shows a trend
toward positive SWBTD as TPW increases, owing to enhanced 12.3 µm absorption.
The effect of increasing TPW on the structure of the dust signal in Fig. 11 follows this same positive trend, with little
variation in dust layer height for low-AOD layers. However, as the dust layer AOD increases, two principal effects are
noted. The first effect is the increasing spread of SWBTD values, for a given dust AOD and TPW value, among a family of
dust layer heights. The lower altitude layers are less negative than the more elevated layers, as the former reside below a
deeper column of atmospheric moisture and thus experience greater suppression of the negative SWBTD dust signal. The
second effect is, for a given dust AOD and for increasing TPW, the divergence of SWBTD for the family of dust layer
heights. This behavior appears for AOD > 0.5, and is most prominent for high values of AOD. Here, very little impact of
water vapor is seen for dust layers in the middle to upper atmosphere, while strong, non-linear impacts occur for the low-
altitude dust. The spread of SWBTD within a family of dust layer heights over the range of TPW shown varies from a few
degrees to ten degrees, with values of 5-7 kelvin for an optically thicker (AOD 0.5 to 2.0; e.g., Fig. 4b) dust plume in the
lower atmosphere.



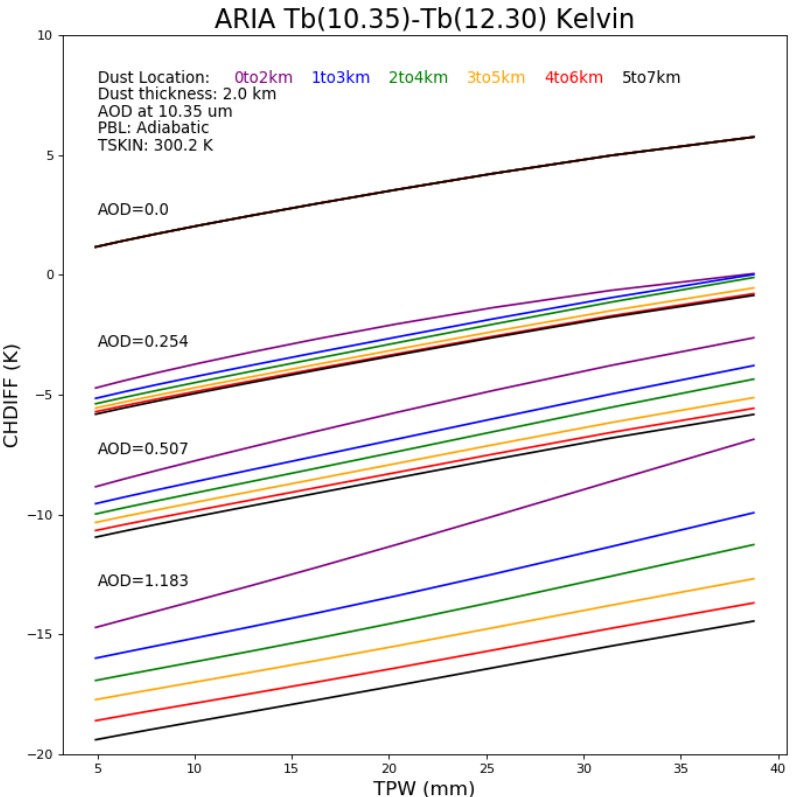

**Figure 11: SWBTD (CHDIFF) for clear-sky conditions and for various configurations of dust layer optical thickness and dust**

**height (ARIA dust type), shown as a function of column-integrated atmospheric moisture (TPW).**

Figure 12 compares the RTM analyses for ARIA, OPAC, and DB17 RI databases assuming the same effective dust particle

radius, loading, and atmospheric profile assumptions as Fig. 11. The expected sensitivity of the SWBTD signal to dust layer

height is evident in all three databases, with higher-altitude dust layers invariably producing stronger (i.e., more negative)

SWBTD dust signals. Also, the higher water vapor concentration of the moist profiles (TPW > 30 mm) result in weaker

(i.e., less negative, or in some cases, positive) SWBTD dust signals compared to the dry (TPW < 20 mm) profiles. The role

of AOD is also evident: since the RI of DB17 yields a much lower 10.35 µm AOD, its associated SWBTD dust signals were

weaker (less negative) than the other two databases. The ARIA database, based on a nearly pure quartz composition,

produced the largest SWBTD dust signals. Interestingly, DB17 exhibits a greater spread among the dust layer altitudes for

higher values of AOD (Fig. 12h,i). The more scattering nature of DB17 extinction at 12.30 µm compared to the other





databases ($\omega_o$ in Table 1) may produce a radiometrically cooler dust layer temperature than more absorbing dust layers—a
signal that is unmasked from the overlying water vapor as the dust layer's altitude is increased.



**Figure 12: Same as Fig. 11 but comparing the ARIA (a-c), OPAC (d-f) and Di Biagio (2017) (g-i) refractive index databases for**
**common assumptions of dust loading.**


Despite the differences in magnitude of signal among the three representative dust types considered, the trends in SWBTD
associated with dust loading, atmospheric profiles, and the heights of the dust layers, were similar in the dry atmosphere.
Thus, whereas variation in the dust mineralogy cannot be neglected when considering selection of SWBTD thresholds, it
cannot explain the lack of a negative SWBTD dust signal as was observed in the 4 August 2016 missing plume case,
particularly for the optically thick component of the missing plume residing over inland portions of the UAE. Thus, the role
of water vapor must be considered for these IR-based dust detection techniques.
**4.3 Simulated Impacts of Water Vapor**
One practical way to illustrate the impact of water vapor on SWBTD-based dust detection under more realistic (non-
idealized) conditions is via radiative transfer simulations of the SWBTD conducted on the fully-configurable environmental
state of a forecast model. Specifically, we can examine the differential signal for dust as would be observed by satellite for
spatially varying lofted dust within an atmosphere with and without water vapor. A WRF-Chem forecast was used for this
exercise, valid for 1200 UTC on 4 August 2016. Whereas the model does not capture the exact details of the dust and
moisture distributions as observed, it does represent the dry- and moist-embedded dust plumes to an extent that is sufficient
for the analysis of water vapor impacts. Fig. 10 demonstrates the representativeness of the model's dust distribution for this
case. For the radiative transfer calculations, again using the hybrid Eddington model of Deeter and Evans (1998) and
following its implementation by Grasso and Greenwald (2004), we used dust concentration and spatial distribution as
determined by WRF-Chem, and assigned dust optical properties based on the ARIA database described in Sect. 4.2.

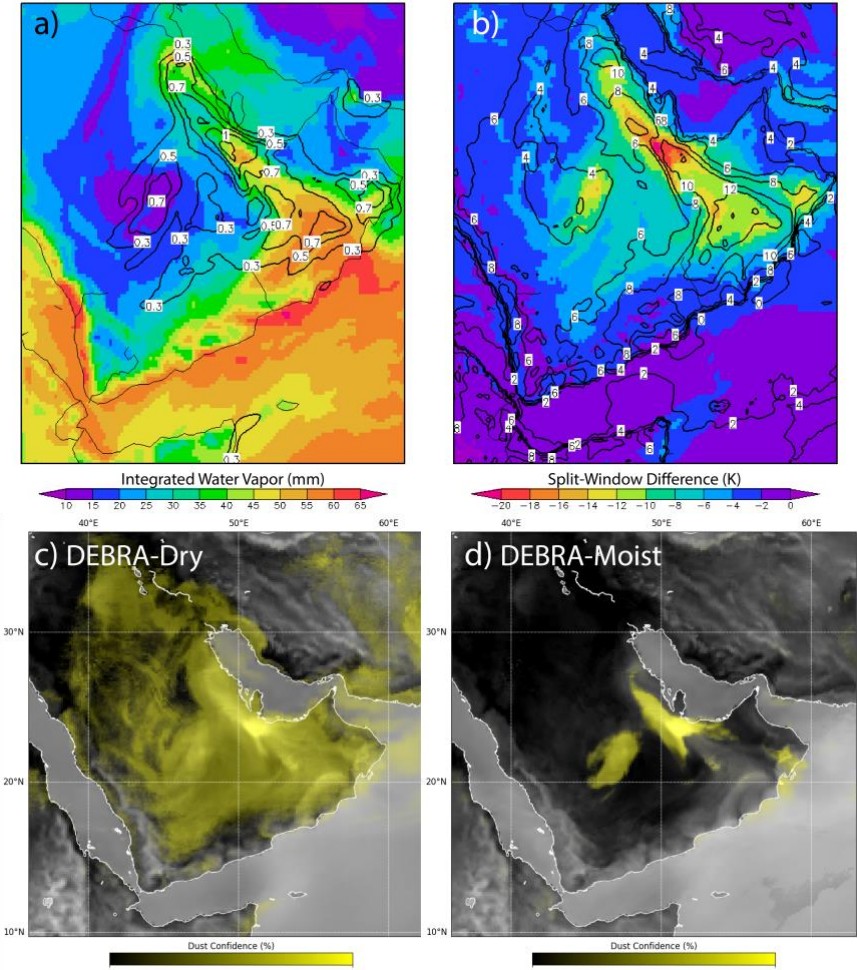

**Figure 13: Model-based analysis of water vapor impacts to SWBTD-based dust detection, based on a WRF-Chem simulation valid at 1200 UTC on 4 August 2016. Panels show a) shaded IWV overlaid by contours of aerosol optical depth at 10.35 μm, b) shaded SWBTD for a dry atmosphere overlaid by contours of SWBTD differences for moist-minus dry atmospheres, c) synthetic DEBRA imagery for a dry atmosphere, and d) corresponding synthetic DEBRA imagery for the full IWV field.**

The results of these simulations are shown in Figure 13. Fig 13a shows the distribution of IWV in shading, and the location of dust plumes, in terms of the 10.35 μm AOD, as contours. Dust in the 1.4 μm WRF-Chem model size bin was used, as it provided the closest match to the DEBRA-based satellite observations of dust distribution for this case. The two significant dust plumes are evident in the contours of Fig. 13a, although the model's moist air mass dust plume is displaced to the south and west of where it was observed (this shift is of no consequence for the illustration of water vapor impacts). Fig. 13b shows in shading the SWBTD dust signal for a 'dry' (i.e., water vapor mixing ratios were set to vanishingly small values of





1.0e-8 g/kg at all levels) atmosphere. The differential effect of moisture on this signal, shown as contours in Fig. 13b, is
defined as SWBTD(moist) minus SWBTD(dry). In this case, 'moist' pertains to the original distribution of atmospheric
moisture in the model. Positive values of this difference show how column moisture is making the SWBTD less negative,
and thus weakening the observable dust signal that is used for dust enhancement by algorithms such as DEBRA.
The impacts to DEBRA can be evaluated directly, by producing a suite of synthetic observations and running these through
the DEBRA algorithm. Based on this approach, Fig. 13c shows how DEBRA would perform in a completely dry
environment, and Fig. 13d shows the corresponding effects of the moisture distribution. While parts of the two significant
dust plumes remain enhanced in the full-atmosphere simulation, the preferential suppression of the dust signal in moist
regions is very evident. This synthetic DEBRA performance is consistent with the idealized single-column simulations of
Sect. 4.2, and its structure anticipated by the location of significant positive-valued contours in Fig 13b. These simulated
results are also consistent to first order with the observed disparity in the IR-based dust detection.
**5 A Vapor-Indexed Dust Detection Method**
Considering the modeled sensitivity of the SWBTD dust signal to column water vapor and the location of the dust in the
profile, we examined to what extent the detection might be improved by incorporating atmospheric column moisture as *a*
*priori* information into SWBTD-based detection algorithms. Whereas the DEBRA dust enhancement enlists a dynamic
lower boundary (accounting for a spatially and temporally varying land surface emissivity signal) for its scaling of the
SWBTD, its upper boundary is held fixed. Moreover, both scaling bounds are predicated on an implicit assumption of a
characteristic or climatological column moisture value (monthly means, computed over several years of observations).
Considering these assumptions, it is not surprising that DEBRA would struggle when confronted with a situation of
anomalous moisture, such was the case in the current study.
To examine the potential of moisture information to improve SWBTD dust detection performance, we enlisted the self-
contained multi-sensor observing system of VIIRS, CrIS and ATMS (the latter two sensors providing the NUCAPS
retrievals) on Suomi-NPP. First, NUCAPS surface-to-500mb IWV data (shown in Fig. 9) were remapped to the VIIRS
domain (Fig. 4) via a bi-linear interpolation. To provide a first-order index for modulating the SWBTD thresholds, these
IWV data were then normalized between low and high bounds of 25 and 45 mm (i.e., set to 0.0 below 25 mm, 1.0 above 45
mm, and ranging linearly in between), respectively. Based on the simulated dynamic range of SWBTD for moderate to
optically thick dust in the lower atmosphere (e.g., Fig. 11; AOD=1.183, 0-2 km), we selected an additive shift factor of
magnitude 7 kelvins. This magnitude was multiplied with the IWV normalized term such that zero shift to the SWBTD
threshold was applied to the low-bound range of IWV, a maximum positive shift of 7 to the high-bound IWV, and variable
shift magnitudes varying linearly for 0 to 7 in between.





The shift factor was introduced to a modified version of the DEBRA algorithm as a first-order correction to the SWBTD dust
signal dampening effect of IWV—applied on a per-pixel basis (i.e., spatially resolved) to both the cloud mask restoral and
the SWBTD dust detection tests.  The results of this procedure, applied to the Suomi-NPP 0921 UTC observations of the 4
August 2016 case, are shown in Figure 14.  Fig. 14a is a reproduction of the original DEBRA dust enhancement (Fig. 4a),
Fig. 14b shows the remapped NUCAPS surface-to-500mb IWV (Fig. 9a), and Fig. 14c is the modified DEBRA result.

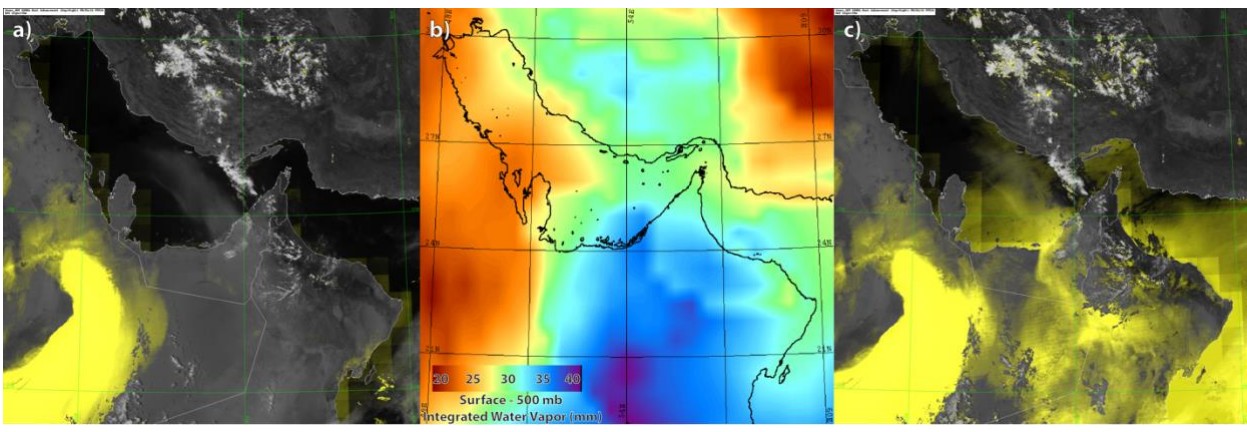

**Figure 14: a) Original DEBRA (Fig. 4a) dust enhancement, b) NUCAPS surface to 500 mb IWV (via Fig. 9a), used as an index to**
**modulate the SWBTD, and c) the revised DEBRA dust enhancement with IWV-modulated SWBTD thresholds applied, capturing**
**significant portions of the missing dust plume both on and offshore.**
Evident in Fig. 14c are portions of the previously 'missing' dust storm over eastern portions of the southeastern Arabian
Peninsula region.  While the enhancements follow the general structure of the elevated IWV region, some areas nested
therein remain unenhanced by DEBRA while other areas appear enhanced above and beyond what the structure of the linear
IWV shift would have suggested.  The region of south-central Iran, for example, remains unenhanced (except for some
cloud-edge artifacts).  Also, the over-land portion of the missing plume along the north-central coastal region of the UAE
shows a clear demarcation between the inland dust and relatively clear air to the west.  Newly-enhanced dust over the eastern
UAE and northern Oman agree with regions discernable from true color (Fig. 4a).  The divergent patterns between IWV and
enhancement give confidence that the IWV shift is not simply imparting an image of itself upon the newly enhanced dust.
Over-water performance, inherently more difficult for SWBTD techniques, is seen to be mixed in Fig. 14c.  The region of
dust over the SG is detected, but not well-isolated, in the modified DEBRA.  The over-water enhancements follow more of
the IWV shift structure, although the presence of dust in these overwater regions is not likely to be false alarm. A lidar
overpass (not shown) from CATS from 4 August which did not cross SG but transected a region just to the east, in the
north/central Gulf of Oman, confirmed the presence of widespread background dust from the surface to 5 km, reminiscent of





the structures seen in Figs. 5 and 6. Thus, the structure of enhancement in over-water dusty regions may follow as an artifact
of the IWV shift pattern, with modulations therein tied to variations of dust optical depth and altitude.
Further improvements to the performance of the IWV-indexed SWBTD dust detection method would require modulation of
the IWV shift factor as a function of an assumed dust vertical profile. If limited-coverage lidar information were available,
from satellite, surface, or aerial platforms, it could be used provide extrapolate a first-guess of this distribution across a
region (e.g., Miller et al., 2014). Alternatively, model information may provide an estimate of the levels in which dust is
most likely to reside. Likewise, in the absence of simultaneous satellite retrievals (e.g., NUCAPS) of column IWV, model
information may also be used to provide a best guess at the atmospheric moisture profile. Data assimilation methods which
consider the covariance between dust and other environmental state parameters (e.g., Zupanski et al., 2018) may also be
enlisted to refine this method.
**5 Summary and Conclusion**
The synergy of numerical modeling, multi-sensor satellite observations, and radiative transfer calculations for varying
aerosol micro- and macrophysical properties offers a unique, multidimensional perspective on a challenging Middle Eastern
dust event. Collectively, they tell a tale of two dust storms embedded within and responding to the dynamics of their parent
air masses—one leading to '*the best of times*' and the other to '*the worst of times*' in terms of SWBTD dust detection
performance. Based on this multi-component analysis, we arrive at the following salient conclusions:
1. Despite the dynamic lower-bound of SWBTD threshold used in DEBRA, which mitigates terrain false-alarm effects,

20        the absolute bounds and range of SWBTD scaling are predicated on an implicit climatological column water vapor

21        assumption. This condition leads to sub-optimal or out-right failure of dust detection in situations of anomalously

22        high moisture that depart from those assumptions.

2. The unique topography of the southeastern Arabian Peninsula and surroundings provides for one such anomalous air

24        mass environment: a southerly surge, capable of lofting significant dust via a variety of mechanisms. These sources

are associated within a maritime (moist) air mass from the northern Arabian Sea.

3. The effects of the moist air mass on the SWBTD are to impart a positive bias which depends on the amount of water

vapor above the dust layer (and hence, the altitude of the dust layer), as well as the lapse rate and surface

temperature. Idealized and model simulations for moist and dry characteristic air masses indicate a spread of

possible SWBTD based on dust composition assumptions, and provide a basis for introducing a modulation to the

SWBTD indexed to the surface-to-500 mb column water vapor.

4. Based on these simulations and the incorporation of NUCAPS IWV information, a new approach to modulating the

DEBRA SWBTD dust detection logic was proposed and demonstrated—yielding improved detection for a dust



1 plume embedded in deep column moisture. Results show promising results for over-land plumes, and mixed

2 performance over water, and suggest the need for further constraints on the vertical distribution of dust.

4 Overall, this study underscores the important roles of atmospheric moisture and dust vertical structure in satellite-based

5 SWBTD detectability of lofted mineral dust. Demonstrated here for the Arabian Peninsula, these challenges are germane to

6 other regions of the world—wherever arid/semi-arid regimes juxtapose with tropical/maritime humid air masses. At such

7 interfaces, intrusions of anomalous air masses will influence SWBTD performance. This study demonstrates that with a

8 priori information on the moisture profile and dust altitude, a limited ability to improve the analysis of dust is possible.

10 This research finds promise in the incorporation of water vapor-indexed information, either measured or modeled, into the

11 SWBTD dust detection techniques. As such, these results point to the benefits of multi-sensor applications and model

12 fusion. Whenever possible, combining IR with visible and/or active sensor (when available) information will reduce

13 detection ambiguity over both land and water backgrounds. If active sensor information is available even for a local cross

14 section (e.g., as provided by CALIPSO or CATS curtain observations), constraints on aerosol layer height within a given

15 environment would enable a more dynamic SWBTD algorithm that could yield further improvements to dust detection. *A

16 priori* information on aerosol layer heights could also be provided by numerical modeling, which can incorporate

17 intermittent past observations on the aerosol profile (e.g., from lidar) and carry that information forward to the current

18 observation time via the model forecast.

20 Additional papers exploring diverse topics, including model-based aerosol distribution/property sensitivity analyses

21 (Bukowski et al., 2018; Saleeby et al., 2018), new three-dimensional model visualization approaches (Albers et al., 2018),

22 and coupled data assimilation for improved analysis and forecasting (Zupanski et al., 2018; Wu et al., 2018), augment the

23 HAALE-MURI research conducted for this case study. Interested researchers are encouraged to contact the authors of this

24 paper and others of the ACP/AMT Special Issue to collaborate on aerosol remote sensing techniques, process

25 characterization and forecasting topics of mutual interest.

26 **Author Contributions**

27 SM conceived the study based on an event first identified by JR, and prepared the manuscript with the help of all co-authors.

28 LG, QB, SK, XX, and CC supported radiative transfer calculations and dust optical property analyses. JD, JS, YW, and JW

29 provided satellite data, tools, and analysis. AW conducted the synoptic scale analysis, and JB, SV, and MZ provided model

30 data supporting case interpretation.



**Acknowledgments**
The support of the Office of Naval Research under grant N00014-16-1-2040 and the NOAA JPSS Program Office are
gratefully acknowledged.

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
