# Peer review of "A Tale of Two Dust Storms: Analysis of a Complex Dust Event in the"

_Atmospheric Measurement Techniques, 2019_

## Referee Comment (RC1) · Michael Garay (Referee) · 19 May 2019

This paper investigates two dust storms that occurred on 3-4 August 2016 and the effect that the environment had on the ability of a particular type of remote sensing algorithm to detect them. The focus is on the infrared split window brightness temperature difference (SWBTD) technique, which relies on differences in the absorption features of silica containing minerals at 10 and 12 $\mu$m, which leads to greater extinction at 10 $\mu$m compared to 12 $\mu$m. Water vapor has the opposite behavior, which greater extinction at 12 $\mu$m compared to 10 $\mu$m, meaning that the presence of significant water vapor could potentially obscure the signal used in the SWBTD technique. The case considered here includes two distinct dust plumes that are readily apparent in visible satellite imagery of the scene. However, two different infrared dust detection algorithms are

both only able to detect the westernmost plume, while missing the easternmost plume nearly completely. Through careful analysis of the meteorological situation and use of additional satellite data sets, the authors convincingly show that the underlying cause of the missed detection is the presence of climatologically elevated amounts of water vapor in the eastern plume. In order to better understand the reasons for the failure of the SWBTD in this situation, the authors conduct a thorough investigation of the physical mechanisms that underpin the SWBTD approach, beginning with the assumed extinction and scattering properties of atmospheric dust. Their results again point to water vapor as the primary culprit in preventing successful retrievals from the SWBTD technique. By including a water vapor correction factor, they demonstrate a path to improving the retrieval in situations with elevated water vapor and make suggestions as to how this can be accomplished in at least a semi-operational manner.

The paper is interesting and extremely well written. For the most part, the analysis is careful and the results are convincing. I find that is paper is appropriate for publication in the journal, after some issues, described below, are addressed. These are primarily of a technical nature and either involve some details that need to be corrected or topics that should be discussed in greater depth.

**Major Comments:**

The major comments are organized by section of the text and include line numbers where appropriate.

Figure 1: It would help to have the two plumes of interested labeled as "1" and "2", not just the missed ("M") portion of one of the plumes.

Figure 4: This would seem to be a prime place to show an image of the SWBTD itself used in the DEBRA retrieval, but this is not done. This would surely be more convincing than the later figure (Figure 5c) that shows the SWBTD, but on the edge of the AIRS swath. As a reader, I find this frustrating because the authors go ahead and check the cloud screening, which to my eye clearly is not the issue, but fail to show what is likely

the root cause of the lack of detection.

Figure 5: In Figure 5c, the colors used to label the locations of "1" and "2" are valid colors in the color scale. Given the need to include these locations in Figure 5a and 5b for reference, I would suggest changing to a different color (like black) since they are labelled anyway, so the color information is redundant. In Figure 5d the color helps, but it does not need to map back to the color of the points because, again, things are labelled. Additionally, the AOD from Deep Blue is shown in Figure 5b, but there is no discussion in the text regarding AOD.

Page 12: When discussing Figure 5 both the AIRS spectra and SWBTD values are shown. It is not clear how the SWBTD is calculated. The header in Figure 5c says "BT difference between 10.35 and 12.3 $\mu$m [K]." Is a "single" wavelength used or some sort of spectral range? It would also help to indicate on Figure 5d where these values were extracted.

Section 3.4: I am a little surprised not to see the AIRS TPW make an appearance in this section. The NUCAPS results are convincing, so maybe the analysis would be redundant. With regard to discussion of the NUCAPS results, the authors write, "the NUCAPS profiles show significant differences in low/mid-tropospheric moisture..." It would be useful to quantify these differences. For example, you can read off the water vapor mixing ratio on the skew-T at 500 hPa and use that information to quantify the difference (it is a little hard to do with the resolution of the figure in the manuscript, however). A point of comparison for a "significant" difference would also be helpful.

Table 1: I happen to have the OPAC numbers easily available, and I see an issue with the way the values are presented in this table. Taking the imaginary part of the index of refraction, it turns out to be -0.5 at both 10 $\mu$m and 12.5 $\mu$m in the OPAC table. However, the 10 $\mu$m value is on a slope, while the 12.5 $\mu$m is at an inflection point in a plot of the imaginary part of the refractive index as a function of wavelength. Given the sensitivity of the calculation of SSA, for example, to this parameter, it is probably

not acceptable just to take the nearest tabulated value. Moreover, the OPAC database provides refractive indices at 10.6 $\mu$m, which is closer to 10.35 $\mu$m than 10 $\mu$m. Here the value for the imaginary part is -0.25, which is a factor of 2 different than the -0.5 value that appears in Table 1. Interpolating using a linear fit to the actual wavelength yields values for the imaginary part of the refractive index of -0.35 for 10.35 $\mu$m and -0.47 for 12.30 $\mu$m. This means that the value is 30% lower at 10.35 $\mu$m and only 6% lower at 12.30 $\mu$m, which may affect the conclusions the authors make based on the dust optical properties alone. It is unclear how carefully the other numbers derived from the ARIA and DB17 databases were calculated, but similar errors might be expected.

Page 20: It is fine to mention that "the spherical particle approximation of Mie theory was assumed in computing the dust optical properties," but the question is how much error (approximately) is made with this assumption given that dust is non-spherical. This is treated in some detail in the infrared by Klüser et al. (2016).

Table 2: There seems to be an issue with the AOD values at 12.30 $\mu$m reported for DB17. The AOD should be proportional to the dust loading, so a plot of the AOD against the dust loading should be a line. For all other dust types, including the ones at 10.35 $\mu$m, this relationship holds, but DB17 has a "kink." My suspicion is that the AOD entry for a dust loading of 186 $\mu$g/kg should really be 0.591, not 0.291 as reported in the table.

Page 21: The authors comment, "Despite these differences, the proportional relationship of extinction between the two wavelengths does not change among the various composition assumptions." Is it the proportion or the difference that matters for SWBTD? Taking the middle row from Table 2, the ratio (proportion) of 10.35/12.30 $\mu$m is 3.8, 1.58, 1.25 for ARIA, OPAC, and DB17, respectively. This is a difference of up to a factor of three. Based on the note above regarding Table 1, these differences might be even larger if more appropriate refractive indices were used. The authors then comment, "hence, the sign of the SWBTD for mineral dust remains negative for all three databases." This statement is not supported in the analysis shown to this point in

the paper as both the AOD and the "blackbody" curve of the surface temperature both the SWBTD, at least to first order. In fact, Figures 11 and 12 (on the next two pages) demonstrate this exact point. I think it would more correct to state at this point in the paper, "hence, the sign of the SWBTD for mineral dust for extinction alone remains negative for all three databases."

Page 26: With regard to the DEBRA results presented in Figure 13, there is no discussion as to why the "dry" case reports so much dust compared to the "moist" case. While it is clear that the increase in TPW results in fewer plumes being identified as dust in the "moist" case, it appears that nearly everything, with the exception of what I assume are cloud fields, is identified as dust in the "dry" case. The explanation is obviously related to the thresholds used in DEBRA, but some discussion of this would be appropriate at the end of this section.

Page 28: At the end of this section the authors argue that models could provide some of the missing information needed to refine the vapor-indexed dust detection method. For example, models can provide vertical information on the location of the dust or a best guess for the atmospheric moisture profile. However, on page 24, Lines 12-14, the authors write, "whereas the [WRF-Chem forecast] model does not capture the exact details of the dust and moisture distributions as observed, it does represent the dry- and moist-embedded dust plumes to an extent that is sufficient for the analysis of water vapor impacts." Given that WRF-Chem runs are already computationally expensive, and do not provide sufficient detail on the observed dust and moisture distributions, is it really likely that other model runs, even with assimilation, would be able to provide the information with the necessary level of fidelity to really impact the results? Proving this point one way or another is clearly beyond the scope of this paper, but I feel the statements made at the end of this section should be somewhat more qualified.

Page 29. The authors write, "this study demonstrates that with a priori information on the moisture profile and dust altitude. . ." However, I was under the impression that section 5 only investigated the effect of the moisture profile. In fact, in the introduction to this section the authors state specifically, "... we examined to what extent the detection might be improved by incorporating atmospheric column moisture as *a priori* information into SWBTD-based detection algorithms."

**Minor Comments:**

Minor comments are provided mainly as suggestions to the author. Line numbers are provided where appropriate.

Page 1, Line 26: Missing an article. "... indexed to an independent-sensor..."

Page 2, Lines 10-11: On page 1, "littoral zone" is presented as a synonym for "coastal zone," this sentence asks us to consider "littoral zone aerosol properties... in coastal zones..." The terms are used somewhat interchangeably throughout the paper, but it does not make sense here. It would be cleaner to write "...characterization of aerosol properties for short-term forecasting applications in coastal zones..."

Page 3, Line 18: O2 should be subscripted $O_2$

Page 3, Line 20: The citation should probably be just "(Xu et al., 2017; 2018)"

Page 3, Line 22: The literature is unclear if "Reststrahlen" should be capitalized. I would argue that since it is just a German word (and not the name of someone), it should not be capitalized. In fact, on the next page it is not capitalized. Also, I think the parentheses in this sentence are not quite what is intended. It should read something like: "...involve the restrahlen band of silica (or quartz), a common and often significant constituent of mineral dusts found worldwide (Di Biagio et al., 2017), caused by..."

Page 4, Lines 3-4: I do not think this sentence refers to the "spectral band." Instead, it would more clearly be the "atmospheric window" that is being discussed.

Page 4, Lines 11-12: I think you need to lead the reader through the "working hypothesis" that you are proposing. Because the signal for water vapor has the opposite sign for mineral dust in the SWBTD, in the presence of both significant water vapor

and atmospheric dust, the signals might cancel one another, leading to a violation of an expected BTD threshold and, consequently, the lack of detection for the case in question.

Page 4, Line 18: Strictly speaking, I would say that the lapse rate of the lower atmosphere over the desert in the daytime is "close to dry adiabatic."

Page 5, Line 4: Is the reference "Tramutoli, 2005, 2007" or "Tramutoli et al., 2005, 2007"? The reference list suggests it should be the former.

Page 7, Line 5: Should be "Cotton et al., 2003"

Page 9: Lines 3-4: probably ". . . a major challenge for numerical modeling. . ."

Page 10, Line 13: There is no reference for the Miller et al. (2009) citation in the reference list.

Page 11, Line 9: Seems like there should be a reference to the MODIS Deep Blue algorithm.

Page 13, Line 7: It is always confusing, but I still call it the "CALIPSO satellite" so it should probably be ". . . on the NASA Cloud-Aerosol Lidar and Infrared Pathfinder Satellite Observation (CALIPSO) satellite (Winker et al., 2009). . ."

Page 14, Line 2: Up until now, the CALIOP lidar and the CALIPSO satellite were referred to correctly. The backscatter shown in the figure is properly from "CALIOP" and not "CALIPSO."

Page 14, Line 8: ". . .CATS and CALIOP profiles. . ."

Page 16, Line 6: ". . .CATS and CALIOP observations. . ."

Page 16, Line 9: "...seen in the CALIOP data. . ."

Page 16, Line 15: "...CATS and CALIOP would. . ."

Page 18, Line 5: Should be "Nalli et al. (2016)"

Page 21, Line 11: I assume this is a "dry adiabatic temperature profile..."

Page 21, Line 21: "Kelvin" should be capitalized.

Page 23, Line 2: The phrase is typically "unmasked by" rather than "unmasked from" as it appears here. The meaning of the sentence is that this effect is "...a signal that becomes increasingly apparent as the influence of the overlying water vapor decreases as the dust layer's altitude is increased."

Page 24, Lines 3-4: "... it alone cannot explain..."

Page 24, Line 10: "...observed by a satellite for..."

Page 25, Line 9: "...observations of the dust distribution..."

Page 26, Line 1: Should check with the journal how this small number should be represented.

Page 26, Line 30: "7 Kelvin."

Page 27, Line 19: To clear up confusion, the sentence could conclude, "... is not simply imparting an image of itself upon the newly enhanced dust over land."

Page 28, Line 6: Maybe what is meant is, "... it could be used to provide an extrapolated first-guess..."

Page 28, Line 12: I think "Summary and Conclusion" should be Section 6.

Page 28, Line 25: I think it should be either "associated with" or "located within" "a maritime (moist) air mass..."

Page 29, Lines 7-8: "a priori" should be italicized.

Page 29, Lines 21-22: The Albers et al. reference is listed as "submitted," but here it appears as 2018, and the same is true of the Bukowski et al. reference. The Saleeby et al. reference says it was submitted to ACP in 2019, but here it appears as a 2018 reference. The Zupanski et al. and Wu et al. papers at least appear as submitted in

2018 in the reference list, but their actual publication date may be different, of course.

Page 30, Line 13: The date for the Cotton et al. reference appears in the wrong place for this reference style.

Page 30, Line 24: The Ginoux et al. (2011) reference does not appear in the body of the paper.

**Reference**

Klüser, L., Di Biagio, C., Kleiber, P. D., Formenti, P., and Grassian, V. H.: Optical properties of non-spherical desert dust particles in the terrestrial infrared – An asymptotic approximation approach, J. Quant. Spectrosc. Radiat. Transf., 178, 209 – 223, https://doi.org/10.1016/j.jqsrt.2015.11.020, 2016.

---

## Referee Comment (RC2) · Michael Folmer (Referee) · 3 Jun 2019

A Review and Recommendation for "A Tale of Two Dust Storms: Analysis of a Complex Dust Event in the Middle East" by Steven D. Miller and Co-Authors.

It was a pleasure reading this manuscript as the content, organization, and general flow were excellent. It is more meaningful when a manuscript can explain complex topics in a simple, effective manner, guiding the reader on a journey of how the authors' scientific approach was used to enhance or improve observational or other techniques. This manuscript easily achieves this goal and allows the reader to follow along without getting lost in the "weeds".

The goal of this manuscript is to show how differing moisture profiles can have signif-

icant impacts on the detection of dust plumes both over land and water. Forecasters experience these impacts in operations fairly often, and they can attest to how difficult it can be to follow a dust plume, and thus the impacts of said plume on convection or lack thereof. The DEBRA technique and the EUMETSAT Dust RGB both have their strengths and weaknesses, but it's nice to see the challenges of atmospheric moisture above dust getting much deserved attention. The dust event used in this case study was very appropriate as it illustrates the observational sensitivities to local geography coupled with complex atmospheric phenomena. A future paper might explore the challenges of following a dust plume from Africa into the Atlantic Ocean, coupled with a tropical wave as this new technique would be very appealing. The use of the integrated water vapor and NUCAPS soundings is a smart approach along with an explanation of how different aerosols are detected due to scattering. This improvement to the DEBRA algorithm would have a very positive impacts to operations, but it's understandable that further improvements are hindered by the lack of vertical observations (i.e. broader swaths from CATS or CALIPSO), yet modeling is a viable option.

After a few minor revisions, I recommend this manuscript to be accepted.

Minor edit suggestions: Page 2, line 21: Change "importance" to "important" so the sentence flows better into line 22. Page 12, lines 11-12: Re-word "Of note in Fig. 5d is that, whereas Location 3..." This feels like a stumble on the way to trying to understand the SWBTD signal. Page 13, line 4: Insert commas after "observations" and "study" Page 13, line 10: Delete "perhaps" as this seems wordy. Page 17, lines 8-9: Change "By inspection with Figs. 1, 4 and 5, this dry..." to "Close inspection of Figs. 1, 4, and 5, shows dry..." Page 18, line 2: Either change "JPSS-1" to "NOAA-20" or include in the parenthesis (JPSS-1/NOAA-20) as the name changed once it became operational. Page 19, line 8: Add "to" between "due" and "a". Page 28, line 6: Re-word "it could be used provide extrapolate a first-guess..." as this is very confusing. I can't make a suggested correction because I'm not sure what you are trying to say (it could be used to provide or it could be used to extrapolate). Page 29, line 1: Re-word "Results show

promising results. . .” as this is redundant (i.e. Results are promising. . .)

---

## Author Comment (AC1) · 25 Jul 2019

**Review Response**

**"A Tale of Two Dust Storms: Analysis of a Complex Dust Event in the Middle East"**
**by Steven D. Miller et al.**

We would like to thank our two peer Reviewers, Michael Garay and Michael Folmer (both electing to forgo anonymity) for their constructive critiques to our manuscript. We address their questions and suggestions in an itemized way here, and have provided a revised manuscript with changes tracked to show where improvements have been made. We apologize for the delay in returning this revision, due to busy summertime schedules, and appreciate the extra time allotted by the Editor and the patience of our Reviewers during this process.

The Reviewer comments are reproduced below, and our responses to those comments are denoted in blue text.
* * *
**Reviewer 1: Michael Garay**
michael.j.garay@jpl.nasa.gov
Received and Published: 19 May 2019

(Preamble Remarks…)

(…)

I find that is paper is appropriate for publication in the journal, after some issues, described below, are addressed. These are primarily of a technical nature and either involve some details that need to be corrected or topics that should be discussed in greater depth.

Thank you for these comments! We have made our best effort to revise the manuscript per your advice.

**Major Comments:**

Figure 1: It would help to have the two plumes of interested labeled as "1" and "2", not just the missed ("M") portion of one of the plumes.

Good suggestion, the changes to Figure 1 have been made and caption/discussion text modified accordingly.

Figure 4: This would seem to be a prime place to show an image of the SWBTD itself used in the DEBRA retrieval, but this is not done. This would surely be more convincing than the later figure (Figure 5c) that shows the SWBTD, but on the edge of the AIRS swath. As a reader, I find this frustrating because the authors go ahead and check the cloud screening, which to my eye clearly is not the issue, but fail to show what is likely the root cause of the lack of detection.

Frustrating the reader is certainly something we wish to avoid! An additional panel, showing the SWBTD itself, has be included in a revised version of Fig. 4.

Figure 5: In Figure 5c, the colors used to label the locations of "1" and "2" are valid colors in the color scale. Given the need to include these locations in Figure 5a and 5b for reference, I would suggest changing to a different color (like black) since they are labelled anyway, so the color information is redundant. In Figure 5d the color helps, but it does not need to map back to the color of the points because, again, things are labelled. Additionally, the AOD from Deep Blue is shown in Figure 5b, but there is no discussion in the text regarding AOD.

We considered this suggestion and can see your reasoning behind making it. Changing the points to black renders point 3 nearly impossible to see in Fig. 5c, so instead we have made these location points white. This presents a minor lingering ambiguity with respect to the Fig 5c color bar, but we feel that it is far less confusing than the original version (where two colors mapped to the bar), and hope that you will agree with the compromise.

Your comment about AOD (5b) not being used in the text alerted us to a figure referencing error in the main text. Statements included at the end of Section 4.2 were supposed to reference Fig. 5b, but 4b was listed instead. This typo has been corrected in the revision.

Page 12: When discussing Figure 5 both the AIRS spectra and SWBTD values are shown. It is not clear how the SWBTD is calculated. The header in Figure 5c says "BT difference between 10.35 and 12.3 um [K]." Is a "single" wavelength used or some sort of spectral range? It would also help to indicate on Figure 5d where these values were extracted.

Thanks for pointing this out to us. The SWBTD from AIRS is determined from single wavelength (monochromatic) values selected from the AIRS spectra; they are not band averaged. We did examine the effect of band averaging over narrow spectral windows (< 0.5 μm), and although there were minor differences the general behavior is the same. The main purpose of showing Fig. 5 is to give the reader an appreciation for the spectral structure of signal (or lack thereof) for the dry- and moist-air associated dust plumes.

Checking this, we did notice that the parenthetical values reported in Fig. 5d were in fact taken from a 11-12 μm version of the SWBTD (a carryover from a previous draft of the manuscript, before we opted to use the 10.35 and 12.3 μm bands in all of our simulations). For consistency with the rest of the manuscript, we have updated the figure to report the 10.35 -12.3 μm SWBTD values. The samples at locations 1-3 thus change, and although the main point does not change, the values reflect even more dramatic differences between the detected and missed dust plumes since 10.35 μm digs deeper into the reststrahlen band.

Section 3.4: I am a little surprised not to see the AIRS TPW make an appearance in this section. The NUCAPS results are convincing, so maybe the analysis would be redundant. With regard to discussion of the NUCAPS results, the authors write, "*the NUCAPS profiles show significant differences in low/mid-tropospheric moisture…*" It would be useful to quantify these differences. For example, you can read off the water vapor mixing ratio on the skew-T at 500 hPa and use that information to quantify the difference (it is a little hard to do with the resolution of the figure

in the manuscript, however). A point of comparison for a "significant" difference would also be helpful.

While we agree that a discussion that brings in AIRS would be interesting, especially if done in the context of a model analysis and with a focus on the lower atmospheric structure, we felt that doing so here would not add to the discussion (it would be redundant, as you say). Doing justice to this would require something more involved than an excurses in the current paper could accommodate, but it provides a nice idea for a follow-on study. The moist/dry features of interest are consistent with the WRF-Chem TPW and with rawinsondes (available but also not shown in this paper for the interest of brevity). Here, it should also be noted that WRF-Chem is initialized by the Global Data Assimilation System (GDAS), which does assimilate AIRS brightness temperatures as one of the many observational datasets feeding that gridded product. Ultimately, the use of NUCAPS is preferred in this case because we seek to enlist the space/time-matched observational information as an index to a revised version of DEBRA (Section 5).

To the comment of significance, we have added some values to the caption and text related to the surface to 500 mb IWV. Fig. 9a is meant to provide a bulk view of the IWV over this layer, while Figs. 9b and 9c are meant to show the structure of the dewpoint depression in the vertical. We have provided additional detail to the comparison by noting the difference in dewpoint depression values between the dry and moist sounding locations.

Table 1: I happen to have the OPAC numbers easily available, and I see an issue with the way the values are presented in this table. Taking the imaginary part of the index of refraction, it turns out to be -0.5 at both 10 um and 12.5 um in the OPAC table. However, the 10 um value is on a slope, while the 12.5 um is at an inflection point in a plot of the imaginary part of the refractive index as a function of wavelength. Given the sensitivity of the calculation of SSA, for example, to this parameter, it is probably not acceptable just to take the nearest tabulated value. Moreover, the OPAC database provides refractive indices at 10.6 um, which is closer to 10.35 um than 10 um. Here the value for the imaginary part is -0.25, which is a factor of 2 different than the -0.5 value that appears in Table 1. Interpolating using a linear fit to the actual wavelength yields values for the imaginary part of the refractive index of -0.35 for 10.35 um and -0.47 for 12.30 um. This means that the value is 30% lower at 10.35 um and only 6% lower at 12.30 um, which may affect the conclusions the authors make based on the dust optical properties alone. It is unclear how carefully the other numbers derived from the ARIA and DB17 databases were calculated, but similar errors might be expected.

You are correct. The following OPAC values were used to obtain updated values of the index of refraction at 10.35 um and 12.3 um:

| | |
|---|---|
| 10.0 um: | 2.57 + 0.5i |
| 10.6 um: | 1.91 + 0.25i |
| 11.5 um: | 1.81 + 0.35i |
| 12.5 um: | 1.74 + 0.5i. |

As you suggested, linear interpolation yielded the following:

10.35 um:       2.19 + 0.35i
12.3 um:        1.75 + 0.47i.

A new set of figures was computed, replacing the corresponding panels (d-f) of Fig. 12 in the revised manuscript. Provided below is a comparison between the old set and new set. Although differences are non-trivial and the correction was necessary, the conclusions drawn from both sets of results in terms of water vapor impacts on the SWBTD remain unchanged.

We double-checked the ARIA and DB17 databases, and those values were correct as-is.

[Figure]

***Above***: SWBTD simulations based on old dataset

[Figure]

***Above***: SWBTD simulations based on new dataset, using the following OPAC indices of refraction: 10.35 um 2.19 + 0.35i and at 12.3 um 1.75 + 0.47i.

Page 20: It is fine to mention that "*the spherical particle approximation of Mie theory was assumed in computing the dust optical properties*," but the question is how much error (approximately) is made with this assumption given that dust is non-spherical. This is treated in some detail in the infrared by Klüser et al. (2016).

We must proceed with some degree of caution here, as a meaningful comparison between spherical and non-spherical dust is very challenging in general, and specifically so from the currently available data. For example, whereas the index of refraction for the OPAC case at 10.35 μm is 2.19 + 0.35i (see above), the closest values of the index of refraction in the non-spherical database (Meng et al. 2010) at 10.35 μm are 2.1 + 0.2i and 2.1 + 0.5i at 10.35 um. The

40 possible sets of optical properties that each depend on the particle aspect ratio further complicates this problem.

As this Reviewer is well aware, there is considerable variability in the values of dust optical properties. The Klüser et al. (2016) study, among others, provides a nice review of this challenge. We have cited this reference in the revision.

It so happens that an equivalent spherical geometry is included as part of the non-spherical database, such that a direct comparison can at least be made with the single-scatter values derived from Mie theory:

Spherical OPAC 10.35 um:
Index of Refraction = 2.19 + 0.35i:    ke = 0.29714, wo = 0.52594, g = 0.51423

Spherical category available from database at 10.35 um:
Index of Refraction = 2.1 + 0.2i:      ke = 0.24281, wo = 0.597583, g = 0.37913
Index of Refraction = 2.1 + 0.5i:      ke = 0.24952, wo = 0.452224, g = 0.38101

The most significant difference here appears to be in the asymmetry parameter, which is less forward scattering.  And

Instead of going down the rabbit hole of conducting a full assessment of uncertainty tied to these simulations, we have added discussion that speaks to departures from spherical geometry as studied by Klüser et al. (2016), while drawing the additional caveat that pristine species and single scatter assumptions are seldom what are actually encountered in nature (e.g., Baran et al., 2005). To the extent that the assumptions made here provide SWBTD that are representative to those observed, we believe this is the appropriate level to conduct this discussion of relative sensitivity, realizing that the exact magnitude of behavior will vary in nature.  The additional text (in italics) follows:

*The spherical particle approximation of Mie theory was assumed in computing the dust optical properties. Using simulated and measured spectra, Klüser et al. (2016) find significant variance of infrared dust optical properties for spherical vs. aspherical (specific "habits" of needles and disks) and varying minerology and assumptions. Specifically, while the general structure of extinction is similar among the permutations, they find shifts to peak extinction and moderate variation to the structure of single scatter albedo, phase function, and its associated asymmetry parameter. Mixtures of mineral components, randomly-oriented irregular particle shapes which may skew the bulk optical properties toward those of oblate spheroids (as in the case of ice clouds; Baran et al. (2005)), and the phase function smoothing effects of multiple scatter in the optically thick dust media considered here may reduce the magnitude of disparities implied by the idealized simulations.  Suffice to say that Mie theory, or any fixed assumption on dust optical properties, comes attached with uncertainties (Meng et al., 2010). In what remains a highly under-constrained problem, the simulation results presented hereafter should be interpreted as representative of bulk dust properties that are, to first order, consistent with the satellite-observed SWBTD behavior.*

*With these caveats in mind, (…)*

We realize and apologize for the fact that this is not a very satisfying response, but the unfortunate reality is that this remains a highly under-constrained problem.

In addition to Klüser et al. (2016), we have added this Reference:

Meng, Z., P. Yang, G. W. Kattawar, L. Bi, K. N. Liou, I. Laszlo, 2010: Single-scattering Properties of Nonspherical Mineral Dust Aerosols: A Database for Application to Radiative Transfer Calculations, J. of Aerosol Science, 41, 501-512.

Table 2: There seems to be an issue with the AOD values at 12.30 um reported for DB17. The AOD should be proportional to the dust loading, so a plot of the AOD against the dust loading should be a line. For all other dust types, including the ones at 10.35 um, this relationship holds, but DB17 has a "kink." My suspicion is that the AOD entry for a dust loading of 186 ug/kg should really be 0.591, not 0.291 as reported in the table.

Thank you for bringing this to our attention. We found and corrected a typo for DB17 at mass = 80 from 0.208 to 0.125. Again, the latter was a data entry typo (entered wrong cell from another master table), and the correct value was used in the actual simulations.

Also, since the value of the OPAC index of refraction have changed as a result of our interpolations, entries in Table 2 were updated commensurately.

The new contents of Table 2 are as follows:

| | AOD (1035) | | | AOD(1230) | | |
|---|---|---|---|---|---|---|
| Dust Loading | aria | opac | db17 | aria | opac | db17 |
| 40 | 0.253 | 0.221 | 0.130 | 0.067 | 0.157 | 0.062 |
| 80 | 0.507 | 0.443 | 0.260 | 0.133 | 0.313 | 0.125 |
| 186 | 1.183 | 1.033 | 0.606 | 0.311 | 0.730 | 0.291 |

Page 21: The authors comment, "*Despite these differences, the proportional relationship of extinction between the two wavelengths does not change among the various composition assumptions.*" Is it the proportion or the difference that matters for SWBTD? Taking the middle row from Table 2, the ratio (proportion) of 10.35/12.30 um is 3.8, 1.58, 1.25 for ARIA, OPAC, and DB17, respectively. This is a difference of up to a factor of three. Based on the note above regarding Table 1, these differences might be even larger if more appropriate refractive indices were used. The authors then comment, "*hence, the sign of the SWBTD for mineral dust remains negative for all three databases.*" This statement is not supported in the analysis shown to this point in the paper as both the AOD and the "blackbody" curve of the surface temperature both the SWBTD, at least to first order. In fact, Figures 11 and 12 (on the next two pages) demonstrate this exact point. I think it would more correct to state at this point in the paper,

*"hence, the sign of the SWBTD for mineral dust for extinction alone remains negative for all three databases."*

Point taken. While the ratio remains > 1.0 (that was the real emphasis—if it had switched to < 1.0 then that implied reverse SWBTD behavior), it is not correct to say that the proportional relationship does not change. We have added clarification and modified the specific sentence in question as suggested.

Page 26: With regard to the DEBRA results presented in Figure 13, there is no discussion as to why the "dry" case reports so much dust compared to the "moist" case. While it is clear that the increase in TPW results in fewer plumes being identified as dust in the "moist" case, it appears that nearly everything, with the exception of what I assume are cloud fields, is identified as dust in the "dry" case. The explanation is obviously related to the thresholds used in DEBRA, but some discussion of this would be appropriate at the end of this section.

In Figure 13c, the area of non-zero dust confidence is indeed widespread, suggesting the presence of lofted dust across nearly the entire domain. While this may appear as erroneous, it is actually representative of the dust distribution that is being produced by the model. Areas of lower confidence in the DEBRA-enhanced imagery correspond with areas of AOD ~ 0.1. In real-world circumstances, where the atmosphere is relatively (but not completely) dry, the algorithm would indicate that dust is present in areas of elevated AOD.

With that said, the atmosphere used in our radiative transfer calculations, corresponding to the imagery of Fig. 13c, assumes 0% relative humidity conditions. We also assumed perfect knowledge of the surface background, such that any dust signal could be enhanced uniquely. The lack of any water vapor in the column allows substantially lower concentrations of dust to fall within the algorithm's thresholds, and combined with the zero-noise floor, provides classification as dust in the simulated DEBRA imagery.

We have added clarifying text to this section, to emphasize that Fig. 13d considers the same dust distribution as Fig. 13c, but reveals the impacts of "real world" moisture on IR-based dust detection compared to the "idealized" case of a completely dry atmosphere.

Page 28: At the end of this section the authors argue that models could provide some of the missing information needed to refine the vapor-indexed dust detection method. For example, models can provide vertical information on the location of the dust or a best guess for the atmospheric moisture profile. However, on page 24, Lines 12-14, the authors write, "*whereas the [WRF-Chem forecast] model does not capture the exact details of the dust and moisture distributions as observed, it does represent the dry and moist-embedded dust plumes to an extent that is sufficient for the analysis of water vapor impacts.*" Given that WRF-Chem runs are already computationally expensive, and do not provide sufficient detail on the observed dust and moisture distributions, is it really likely that other model runs, even with assimilation, would be able to provide the information with the necessary level of fidelity to really impact the results? Proving this point one way or another is clearly beyond the scope of this paper, but I feel the statements made at the end of this section should be somewhat more qualified.

The statement was not intended to be general, but specific to this case—the model did a good enough job with producing two major dust plumes (one in a dry air mass and the other in a moist air mass) allowing us to illustrate the effects of water vapor on SWBTD dust detection. We have rephrased the sentence to read:

*Whereas the model does not capture the exact details of the moisture and lofted dust distributions for this case, it is sufficiently representative to illustrate the impacts of water vapor on SWBTD-based dust detection.*

Generally speaking, the state of the art in satellite-derived T/Q sounding assimilation is arguably sufficient to represent the moisture distribution, but it is far less likely to capture the correct dust distribution. In practice, if we have some limited *a priori* information on the vertical level of dust altitude (e.g., a lidar cross section, as in the current case study, or by some other informed assumption), then the concept of vapor-indexed SWBTD detection thresholding becomes tractable. This is discussed (and exemplified for our case study) in Section 5.

Page 29. The authors write, "*this study demonstrates that with a priori information on the moisture profile and dust altitude…*" However, I was under the impression that Section 5 only investigated the effect of the moisture profile. In fact, in the introduction to this section the authors state specifically, "*...we examined to what extent the detection might be improved by incorporating atmospheric column moisture as a priori information into SWBTD-based detection algorithms.*"

The indexed approach detailed in Section 5 does enlist *a priori* information on dust altitude, in this case coming from the lidar (CALIPSO and CATS) datasets, Figs. 6-7, which showed that the optically thick dust resided in the 0-3 km region. This information, however, is very limited, and was used here only to select a spread of expected SWBTD suppression values [0 K, 7 K] for the observed IWV range of [25 mm, 45 mm]. This suppression index can in principle vary with expected dust AOD, dust altitude, and IWV conditions, per Fig. 11, and if additional information on dust type is available, then also per Fig. 12.

In practice, specific values for these parameters may never be well constrained (and if they are, then the dust distribution is likely already determined to an extent that the point of the current approach is mute). We have included additional clarification to this section.

**Minor Comments:**

Minor comments are provided mainly as suggestions to the author. Line numbers are provided where appropriate.

Page 1, Line 26: Missing an article. ": : : indexed to an independent-sensor…"
Corrected.

Page 2, Lines 10-11: On page 1, "*littoral zone*" is presented as a synonym for "*coastal zone,*" this sentence asks us to consider "*littoral zone aerosol properties… in coastal zones…*" The terms are used somewhat interchangeably throughout the paper, but it does not make sense here.

It would be cleaner to write "…*characterization of aerosol properties for short-term forecasting applications in coastal zones*…"
Done.

Page 3, Line 18: O2 should be subscripted $O_2$
Done.

Page 3, Line 20: The citation should probably be just "(Xu et al., 2017; 2018)"
Changed.

Page 3, Line 22: The literature is unclear if "Reststrahlen" should be capitalized. I would argue that since it is just a German word (and not the name of someone), it should not be capitalized. In fact, on the next page it is not capitalized. Also, I think the parentheses in this sentence are not quite what is intended. It should read something like: "…*involve the restrahlen band of silica (or quartz), a common and often significant constituent of mineral dusts found worldwide (Di Biagio et al., 2017), caused by*…"
Agree, and thanks for catching the parenthetical error—corrected.

Page 4, Lines 3-4: I do not think this sentence refers to the "spectral band." Instead, it would more clearly be the "atmospheric window" that is being discussed.
Reworded.

Page 4, Lines 11-12: I think you need to lead the reader through the "working hypothesis" that you are proposing. Because the signal for water vapor has the opposite sign for mineral dust in the SWBTD, in the presence of both significant water vapor and atmospheric dust, the signals might cancel one another, leading to a violation of an expected BTD threshold and, consequently, the lack of detection for the case in question.
We have added more text to help walk through the hypothesis.

Page 4, Line 18: Strictly speaking, I would say that the lapse rate of the lower atmosphere over the desert in the daytime is "close to dry adiabatic."
Changed.

Page 5, Line 4: Is the reference "Tramutoli, 2005, 2007" or "Tramutoli et al., 2005, 2007"? The reference list suggests it should be the former.
Corrected to the former.

Page 7, Line 5: Should be "Cotton et al., 2003"
Corrected.

Page 9: Lines 3-4: probably ": : : a major challenge for numerical modeling…"
Corrected.

Page 10, Line 13: There is no reference for the Miller et al. (2009) citation in the reference list.
Should have been (2008)—corrected.

Page 11, Line 9: Seems like there should be a reference to the MODIS Deep Blue algorithm.
Hsu et al. (2004, 2013) references added here.

Page 13, Line 7: It is always confusing, but I still call it the "CALIPSO satellite" so it should probably be "…on the NASA Cloud-Aerosol Lidar and Infrared Pathfinder Satellite Observation (CALIPSO) satellite (Winker et al., 2009)…"
OK, changed.

Page 14, Line 2: Up until now, the CALIOP lidar and the CALIPSO satellite were referred to correctly. The backscatter shown in the figure is properly from "CALIOP" and not "CALIPSO."
Yes, corrected.

Page 14, Line 8: "…CATS and CALIOP profiles…"
Fixed.

Page 16, Line 6: "…CATS and CALIOP observations…"
Fixed.

Page 16, Line 9: "...seen in the CALIOP data…"
Fixed.

Page 16, Line 15: "...CATS and CALIOP would…"
Fixed.

Page 18, Line 5: Should be "Nalli et al. (2016)"
Fixed.

Page 21, Line 11: I assume this is a "dry adiabatic temperature profile…"
Corrected; dry adiabatic.

Page 21, Line 21: "Kelvin" should be capitalized.
Fixed.

Page 23, Line 2: The phrase is typically "unmasked by" rather than "unmasked from" as it appears here. The meaning of the sentence is that this effect is "…*a signal that becomes increasingly apparent as the influence of the overlying water vapor decreases as the dust layer's altitude is increased*."
We have adopted your suggested phrasing.

Page 24, Lines 3-4: ":…it alone cannot explain…"
Fixed.

Page 24, Line 10: "…observed by a satellite for…"
Fixed.

Page 25, Line 9: "…observations of the dust distribution…"
Fixed.

Page 26, Line 1: Should check with the journal how this small number should be represented.
For now, we have changed to $1.0 \times 10^{-8}$ (it can be changed by the copy-editor later if needed).

Page 26, Line 30: "7 Kelvin."
Fixed.

Page 27, Line 19: To clear up confusion, the sentence could conclude, "…is not simply imparting an image of itself upon the newly enhanced dust over land."
As newly enhanced dust occurs over both land and water, we have added clarification to the previous sentence, as well as clarification to the sentence in question.

Page 28, Line 6: Maybe what is meant is, "…it could be used to provide an extrapolated first-guess…"
Changed to "…it could be used to provide a first-guess of the vertical distribution of dust…"

Page 28, Line 12: I think "Summary and Conclusion" should be Section 6.
Corrected.

Page 28, Line 25: I think it should be either "associated with" or "located within" "a maritime (moist) air mass…"
Corrected to "embedded within a maritime (moist) air mass originating from…"

Page 29, Lines 7-8: "a priori" should be italicized.
Done.

Page 29, Lines 21-22: The Albers et al. reference is listed as "submitted," but here it appears as 2018, and the same is true of the Bukowski et al. reference. The Saleeby et al. reference says it was submitted to ACP in 2019, but here it appears as a 2018 reference. The Zupanski et al. and Wu et al. papers at least appear as submitted in 2018 in the reference list, but their actual publication date may be different, of course.
Updated 2018 submissions to 2019. Saleeby paper In Press. Wu paper deleted (lifted and submitted to a D/A focused journal).

Page 30, Line 13: The date for the Cotton et al. reference appears in the wrong place for this reference style.
Reformatted.

Page 30, Line 24: The Ginoux et al. (2011) reference does not appear in the body of the paper.
Carry-over from older version of manuscript…deleted.

**Reference**
Klüser, L., Di Biagio, C., Kleiber, P. D., Formenti, P., and Grassian, V. H.: Optical properties of non-spherical desert dust particles in the terrestrial infrared – An asymptotic approximation approach, J. Quant. Spectrosc. Radiat. Transf., 178, 209 – 223, https://doi.org/10.1016/j.jqsrt.2015.11.020, 2016.

Thanks for this reference—included.

**Reviewer 2:  Michael Folmer**
michael.folmer@noaa.gov
Received and Published: 3 June 2019

(Introductory Remarks)

(…)

After a few minor revisions, I recommend this manuscript to be accepted.

Thank you for reading through our manuscript!

**Minor Edit Suggestions:**

Page 2, line 21: Change "importance" to "important" so the sentence flows better into line 22.
Done.

Page 12, lines 11-12: Re-word "Of note in Fig. 5d is that, whereas Location 3…" This feels like a stumble on the way to trying to understand the SWBTD signal.
We have reworded this statement for added clarity.

Page 13, line 4: Insert commas after "observations" and "study"
We have also reworded this clumsy statement for added clarity.

Page 13, line 10: Delete "perhaps" as this seems wordy.
Deleted.

Page 17, lines 8-9: Change "By inspection with Figs. 1, 4 and 5, this dry…" to "Close inspection of Figs. 1, 4, and 5, shows dry…"
Corrected this sentence.

Page 18, line 2: Either change "JPSS-1" to "NOAA-20" or include in the parenthesis (JPSS-1/NOAA-20) as the name changed once it became operational.
Changed to NOAA-20.

Page 19, line 8: Add "to" between "due" and "a".
Good catch; corrected.

Page 28, line 6: Re-word "*it could be used provide extrapolate a first-guess…*" as this is very confusing. I can't make a suggested correction because I'm not sure what you are trying to say (it could be used to provide or it could be used to extrapolate).
Corrected per Reviewer 1.  The full sentence now reads:

If limited-coverage lidar information were available from satellite, surface, or aerial platforms, it could provide a first-guess of the vertical distribution of dust across a region (e.g., Miller et al., 2014).

Page 29, line 1: Re-word "Results show promising results…" as this is redundant (i.e. Results are promising…)
Re-worded.

Thanks to both Michael Garay and Michael Folmer for helping us to improve this manuscript. We hope that we have addressed these items to your satisfaction, and we are confident that the changes made have dramatically improved the clarity and content of our submission. It should be noted in closing that through the editing and response process some additional minor points of clarification/grammar correction were made to this revised manuscript—but no items related to the content, results or conclusions drawn from them. All changes made to the manuscript are viewable as tracked-changes. Please let us know if there are any further questions.

Best Regards, on Behalf of the Co-Authors,

-steve

Steven D. Miller
Cooperative Institute for Research in the Atmosphere (CIRA)
Colorado State University
Steven.Miller@colostate.edu
(970) 491-8037

---

## Author Comment (AC2) · 25 Jul 2019

Thank you for providing this review of our manuscript!

We have addressed the comments from Reviewer 1 and 2 via a point-by-point response document. This document is attached here as a supplement file.

Per the journal guidance: "After your posts, you have to explicitly finalize the final-response form before you are asked in a separate email to prepare and submit your revised manuscript for peer-review completion and potential final publication in AMT."

Our revised manuscript, with changes tracked, is ready to submit at this time. Ee will do so upon receipt of separate email as noted above.

[Figure]

Please also note the supplement to this comment:
https://www.atmos-meas-tech-discuss.net/amt-2019-82/amt-2019-82-AC2-supplement.pdf

———————————————————

---

## Author Comment (AC3) · 25 Jul 2019

The adjusted manuscript with track changes on is attached.

Please also note the supplement to this comment:
https://www.atmos-meas-tech-discuss.net/amt-2019-82/amt-2019-82-AC3-supplement.zip
* * *

---

## Author Comment (AC4) · 25 Jul 2019

A version of the revised manuscript with track changes on is attached.

Please also note the supplement to this comment:
https://www.atmos-meas-tech-discuss.net/amt-2019-82/amt-2019-82-AC4-supplement.zip
* * *